# Monte Carlo Multi-Feature Baseline Shapley (MMBS): An axiomatic attribution method for fine-grained explanations of image classification networks

**Dirk Elias Schut**                                                    *Dirk.Schut@cwi.nl*

*Computational Imaging Group*
National Research Institute for Mathematics and Computer Science (CWI)

**Robert van Liere**                                                    *Robert.van.Liere@cwi.nl*
*Computational Imaging Group*
National Research Institute for Mathematics and Computer Science (CWI)
*Visualization Cluster*
Eindhoven University of Technology

**Tristan van Leeuwen**                                                 *T.van.Leeuwen@cwi.nl*
*Computational Imaging Group*
National Research Institute for Mathematics and Computer Science (CWI)
*Mathematisch Instituut*
Utrecht University

**Reviewed on OpenReview:** https://openreview.net/forum?id=LLFIcr7zWh

## Abstract

This paper presents the Multi-Feature Baseline Shapley (MBS) attribution method for explaining the outcome of a neural network for a given input. MBS generalizes the Integrated Gradients (IG) and Baseline Shapley (BShap) methods by introducing a step size parameter. When the step size is set to one, MBS equals BShap, and when it is set to the number of features, MBS equals IG. MBS is an axiomatic method, which means that it was designed to satisfy certain axioms (mathematical properties). These axioms ensure that the attribution maps relate to the neural network in appealing ways, for example, by preserving linearity or symmetry. We prove that MBS satisfies eight axioms that are also satisfied by IG and BShap. To quickly approximate MBS, this paper presents the Monte Carlo Multi-Feature Baseline Shapley (MMBS) method, which is an unbiased estimator of MBS. On image classification tasks, we show that MMBS also approximates a Monte Carlo estimate for BShap while being orders of magnitude faster to compute. Furthermore, we compare MMBS to nine configurations of existing attribution methods on three image classification networks trained on either the Fashion MNIST or ImageNet1k dataset. MMBS has the best area under the deletion curve score on all three networks.

## 1 Introduction

What makes a dog a Dalmatian? That is one of the many distinctions that an image classification neural network must learn to achieve a good score on the popular ImageNet1k benchmarking dataset (Russakovsky et al., 2015). Image classification networks trained on ImageNet1k aim to classify images into one thousand classes. For a given input image, they compute a probability for each of these classes. To explain how an image classification network came to its decision, an attribution map (also called a heatmap, importance map, or saliency map) can be calculated. An attribution map provides an indication of how much each feature (pixel or color subpixel) of a given input image contributed to the calculated probability for a given class. There are many methods for calculating attribution maps. Image classification networks are commonly used to distinguish between classes that only show subtle differences, so a fine-grained attribution method is

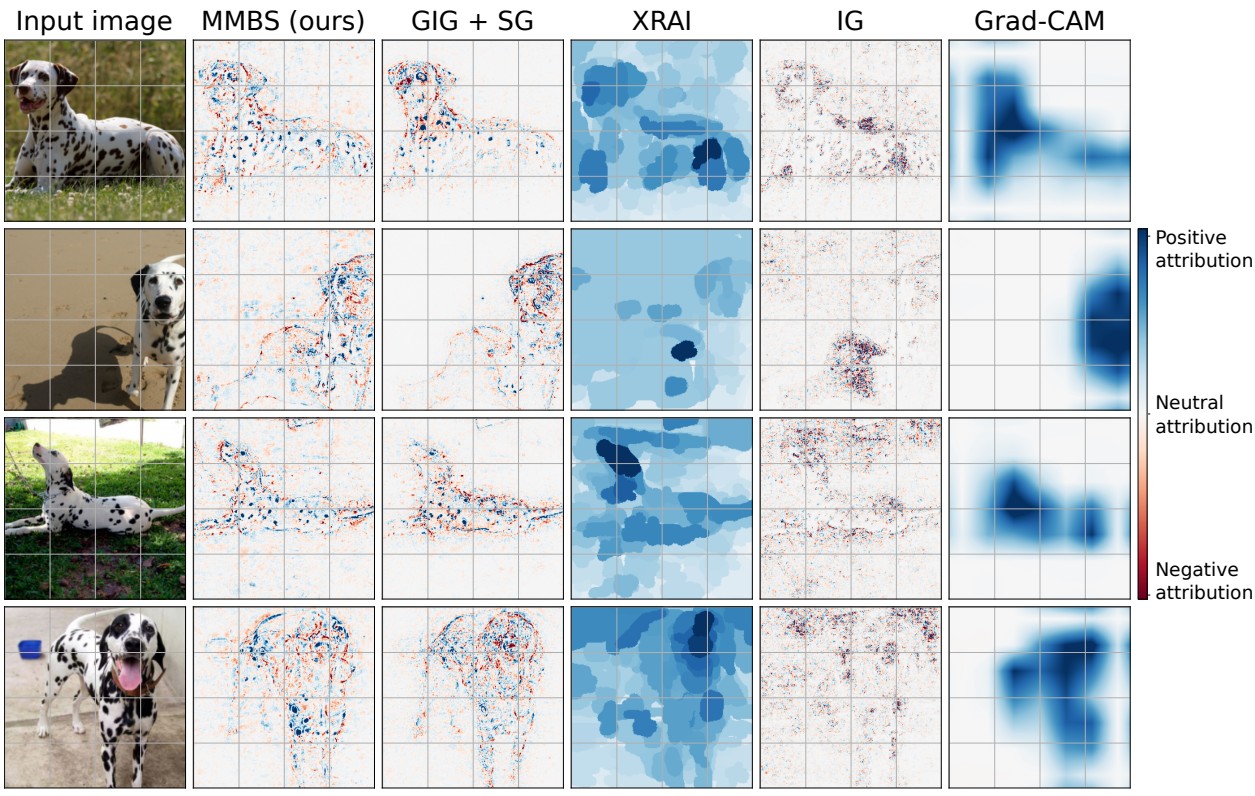

Figure 1: Attribution maps calculated with different methods for the class Dalmatian on images of Dalmatians (IG = Integrated Gradients, GIG + SG = Guided Integrated Gradients with SmoothGrad). There are large differences between the results of different methods.

necessary to highlight these differences. For example, ImageNet1k contains more than one hundred classes of dog breeds. Figure 1 shows the results of several popular methods on images of Dalmatian dogs. There are large differences between the results of these methods.

The weights of a neural network fully specify the network's behavior, so they can be considered an explanation of the network. However, a typical image classification network has millions of weights that are combined in a complex layered structure, making it almost impossible for a human to interpret the weights directly. An attribution map is much simpler for a human to comprehend, because it can be shown as an image. However, an attribution map cannot fully capture a neural network's behavior because it assigns only one score per feature. It is challenging to specify how an attribution map should approximate a neural network's behavior, since it should also explain the network's incorrect or unexpected decisions. Early attribution map methods relied on heuristics that were convenient to calculate (Simonyan et al., 2013; Zhou et al., 2016; Selvaraju et al., 2017). Recently, there has been growing interest in axiomatic attribution methods (Sundararajan et al., 2017; Lundstrom & Razaviyayn, 2025; Sundararajan & Najmi, 2020), which are methods that are designed to satisfy certain axioms (mathematical properties). The axioms define the attribution problem more precisely, and they are chosen so that the attribution map relates to the neural network in an appealing way, for example, by preserving properties of the neural network, such as linearity or symmetry.

Integrated Gradients (IG) is an axiomatic attribution method based on several appealing axioms (Sundararajan et al., 2017; Lundstrom et al., 2022; Lundstrom & Razaviyayn, 2025). However, when applying IG to image classification problems, the resulting attribution maps often have a noisy appearance, with strongly positive and negative attribution values close to each other even within mostly constant image regions, as shown in Figure 1. This makes IG attribution maps challenging to interpret visually. Several papers have proposed modifications of IG to obtain less visually-noisy attribution maps, such as the XRAI (Kapishnikov et al., 2019) and the Guided Integrated Gradients + SmoothGrad (Kapishnikov et al., 2021; Smilkov et al., 2017)

methods, which are also displayed in Figure 1. However, for both methods, it has not been proven that the axiomatic properties of IG are maintained or that additional axioms are satisfied. Baseline Shapley (BShap) is an axiomatic attribution method that satisfies many of the same axioms as IG (Sundararajan & Najmi, 2020). However, the computational cost of BShap grows in the order of $O(n!n)$ with the number of input features $n$, making it impossible in practice to use BShap on images. It is possible to approximate BShap using a Monte Carlo approach (Castro et al., 2009; Mitchell et al., 2022), but that is still computationally expensive, so there is little prior work on what kind of results BShap produces on images. To the knowledge of the authors, it has only been used on very small images (Ancona et al., 2019) or with a very low number of samples (10 samples) (Yeh et al., 2022).

In this paper, we present the Multi-Feature Baseline Shapley (MBS) method. MBS generalizes IG and BShap by introducing a step size parameter (Section 3.2). When the step size is set to one, MBS equals BShap, and when it is set to the number of features, MBS equals IG. We prove that MBS satisfies eight axioms that are satisfied by both IG and BShap (Section 3.3 and Appendix B). We also present the Monte Carlo Multi-Feature Baseline Shapley (MMBS) method, which is an unbiased estimator of MBS (Section 3.4). For image classification on the Fashion MNIST and ImageNet1k datasets, we show that even when the step size is large, MMBS attribution maps are very similar to a Monte Carlo estimate of BShap (Castro et al., 2009; Yeh et al., 2022), but they can be computed orders of magnitude faster (Section 4.2) making it possible to approximate BShap on images for practical use. Moreover, we show that for small, medium (ResNet50), and large (Vision Transformer) neural networks trained on image classification tasks (Fashion MNIST or ImageNet), the MMBS attribution maps achieve lower average area under deletion curve (AUDC) scores than all other state-of-the-art attribution methods we compared it with (Section 4.4).

## 2 Related work

Some of the earlier attribution methods proposed in the literature approximate the behavior of neural networks using heuristics that are convenient to calculate. Simonyan et al. (2013) used the gradient as an attribution map because features with a large gradient should affect the neural network's output most strongly. However, the gradient at a single point is insufficient to capture the full decision-making process due to the highly non-linear nature of neural networks. The Class Activation Map (CAM) (Zhou et al., 2016) and Grad-CAM (Selvaraju et al., 2017) methods use the fact that in convolutional and pooling layers, the inputs and outputs are strongly spatially correlated. This approach can therefore only be applied to convolutional neural networks consisting of many convolutional and pooling layers, with a fully connected block at the end. Moreover, the image size of a Grad-CAM attribution map is equal to the image size of the activations just before the fully connected block, which is typically very small, resulting in a blurry attribution map.

The concept of axiomatic attribution methods was introduced together with the Integrated Gradients (IG) method (Sundararajan et al., 2017). The idea of axiomatic attribution was derived from similar methods in game theory. Specifically, IG is based on the Aumann-Shapley (AS) (Aumann & Shapley, 1974) method for solving the cost-sharing problem in game theory. In IG, a second image, called the baseline image, is used in the calculation. The baseline image should represent a neutral input, and many of the axioms satisfied by IG use it as a reference point. IG attribution maps are calculated by integrating the gradients over the straight-line path between the baseline image and the input image. More rigorous proofs, and proofs for additional axioms have been derived by Lundstrom et al. (2022); Lundstrom & Razaviyayn (2025); Sundararajan & Taly (2018).

IG often produces attribution maps with a noisy appearance. To obtain attribution maps with less visual noise, several modifications have been proposed. Many of these methods modify the integration path. Guided Integrated gradients (GIG) (Kapishnikov et al., 2021) aims to avoid integrating over regions with large unrelated changes in the gradient. To achieve this, the integration path is chosen adaptively so that in each step, only the features with the lowest partial derivatives are modified. SAMP (Zhang et al., 2024) aims to concentrate the attributions to a small number of features. It optimizes the integration path to maximize the variance of the attributions under a Brownian motion assumption. In BlurIG (Xu et al., 2020), the integration path is chosen so that the input image is increasingly blurred in each integration step. In Adversarial Integration (Pan et al., 2021), the integration path is equal to a path that was used to generate an adversarial example (Goodfellow et al., 2014), and the results of multiple of these paths are averaged to

obtain the final result. Important Direction Integrated Gradients (IDGI) (Yang et al., 2023) modifies the integration approach instead of the path, so that in every step along the path, a component in the direction of the local gradient is added. However, none of these modifications of IG have been proven to satisfy new axioms.

The BShap method is another axiomatic attribution method that satisfies many of the same axioms as IG (Sundararajan & Najmi, 2020). It is based on the Shapley-Shubik method (Shubik, 1962; Shapley et al., 1953) for solving the cost-sharing problem in game theory. In this cost-sharing problem setting, the Shapley-Shubik method has been proven to satisfy some axioms that the Aumann-Shapley method, on which IG is based, does not (Friedman & Moulin, 1999; Sprumont, 1998). This suggests that IG and BShap may also have different behaviors on the XAI attribution problem (Sundararajan & Najmi, 2020). However, the use of the BShap method on images has been fairly limited due to the high computational cost (Ancona et al., 2019; Yeh et al., 2022), making it unclear how IG and BShap attribution maps differ when calculated for image classification networks. Similar methods to BShap have been used to calculate attributions for neural networks with fewer input features (Lundberg & Lee, 2017), and variants have been derived for several specific problems, such as trees or linear models (Chen et al., 2023).

Another category of methods takes the output of an existing attribution method (e.g., IG, GradCAM, or IDGI) and modifies it to obtain additional properties, much like adding a regularizer to an optimization problem. XRAI (Kapishnikov et al., 2019) segments locally similar regions into superpixels, and then ranks the attribution of the superpixels based on a (sub)pixel-wise attribution method. SmoothGrad (Smilkov et al., 2017) adds noise to the input image before calculating the attribution map. This is done multiple times with independently sampled noise, and the results are averaged. For many attribution map methods, this results in an attribution map that is visually less noisy. Gildenblat (2021) applied multiple random rotations, scalings (90-110%), and flips, and calculated an attribution map for each transformed version of the input image. The inverse transformations were applied to the results, and these results are averaged to obtain an attribution map that is invariant to these transformations.

While XAI attribution maps remain an active area of research, there are also studies that critique how XAI attribution maps are calculated and used (Marques-Silva, 2023; Kumar et al., 2020). Kaur et al. (2020) showed that data scientists who used attribution maps often misinterpreted them. Moreover, Bilodeau et al. (2024) showed that when an attribution method satisfies the axioms of Linearity and Completeness, it can not do better than random guessing on the *Recourse* and *Spurious feature* tasks. While the MMBS method presented in this paper also satisfies the Linearity and Completeness axioms, it shows a strong performance on the AUDC metric (Petsiuk et al., 2018; Kapishnikov et al., 2019; 2021), showing that, while these axioms limit the performance in some tasks, they do not universally limit the performance across all tasks. In general, we argue that axiomatic guarantees provide valuable information on how an attribution method behaves. Potential users of MMBS, or any other attribution method, should be precise in specifying what they aim to visualize with the attribution map and check whether the method's (axiomatic) properties align with that goal (Bilodeau et al., 2024).

## 3 Methods

In this section, we will first introduce the notation and a formal definition of baseline attribution methods (Section 3.1). After that, we will introduce MBS (Section 3.2) and describe its axioms (Section 3.3). We will then introduce MMBS, a Monte Carlo estimator of MBS (Section 3.4), and discuss its computational cost relative to Monte Carlo estimators of IG and BShap (Section 3.5). Finally, we will describe how the AUDC metric can be used to evaluate attribution map methods (Section 3.6).

### 3.1 Notation and problem formulation

For ease of comparison, we will mostly follow the notation and problem formulation of Lundstrom & Razaviyayn (2025). We denote functions as uppercase letters ($A$), integers and vectors as lowercase letters ($a$), scalars as lowercase Greek letters ($\alpha$), and sets as calligraphic uppercase letters ($\mathcal{A}$). For vectors $a, b \in \mathbb{R}^n$ with $a_i < b_i \,\forall i$ we use the notation $[a, b]$ to denote a subset of $\mathbb{R}^n$ so that $x \in [a, b] \iff a_i \leq x_i \leq b_i \,\forall i$. $\mathcal{F}^2(a, b)$ is the set of neural networks that are composed of real analytic functions and ReLU nonlinearities, that have $[a, b]$ as their domain, and that have a scalar output. Neural networks with multiple outputs can be

modeled within this framework by treating each output as a separate network. For any given $F \in \mathcal{F}^2(a, b)$, its domain is denoted as $\mathcal{D}_F$. We aim to explain a neural network $F \in \mathcal{F}^2(a, b)$ on a certain input (the explicant) $\bar{x} \in \mathcal{D}_F$, relative to some baseline $x' \in \mathcal{D}_F$. Baseline attribution methods are any function in the form $A : [a, b] \times [a, b] \times \mathcal{F}^2(a, b) \to \mathbb{R}^n$. The domain of a given $A$ is denoted as $\mathcal{D}_A$. We also follow some notation from (Friedman & Moulin, 1999): We denote a combination of elements of vectors $x, y \in \mathbb{R}^n$ based on a set $\mathcal{S}$, by $z = x \backslash^{\mathcal{S}} y$ where $z_i = x_i$ if $i \in \mathcal{S}$ and $z_i = y_i$ otherwise. Moreover, we denote the partial derivative of $F$ at $x \in \mathcal{D}_F$ in the $i$-th coordinate as $\partial_i F(x)$. Finally, $\mathcal{R}$ is the set of all possible orderings of the numbers $[1, 2, \ldots, n]$, so it contains $n!$ orderings. If an ordering $r \in \mathcal{R}$ ranks feature $i$ as the first feature and feature $j$ as the second feature, then $r_i = 1$ and $r_j = 2$.

## 3.2 Multi-Feature Baseline Shapley (MBS)

The Multi-Feature Baseline Shapley (MBS) method is calculated by dividing all features in $\bar{x}$ and $x'$ into steps of size $m \in \mathbb{R}$ with $m > 0$ for every possible ordering $r \in \mathcal{R}$. The number of steps is $\frac{n}{m}$ rounded up to the nearest integer $\lceil \frac{n}{m} \rceil$. In every step, an IG attribution map $A^{\text{IG}}$ is calculated, and the values of this attribution map are only non-zero for the features included in that step:

$$A^{\text{MBS}}(\bar{x}, x', F) = \frac{1}{n!} \sum_{r \in \mathcal{R}} \sum_{k=1}^{\lceil \frac{n}{m} \rceil} A^{\text{IG}}(\bar{x} \backslash^{\mathcal{I}(r,k)} x', \bar{x} \backslash^{\mathcal{I}(r,k-1)} x', F), \tag{1}$$
$$\mathcal{I}(r, k) = \{l : r_l \le km\}.$$

$A^{\text{IG}}$ is the IG method, and it is defined as follows (Sundararajan et al., 2017):

$$A_i^{\text{IG}}(\bar{x}, x', F) = (\bar{x}_i - x'_i) \int_{\zeta=0}^1 \partial_i F((1 - \zeta)x' + \zeta\bar{x})d\zeta. \tag{2}$$

MBS is a generalization of the IG method, because when $m = n$ there is only one step for every ordering, in which IG is calculated over all features. This is the same for every ordering, so averaging over all orderings, yields the IG heatmap. Therefore, MBS is equal to IG when $m = n$.

MBS is also a generalization of the BShap method, because MBS equals BShap when $m = 1$. BShap is defined as follows (Sundararajan & Najmi, 2020):

$$A_i^{\text{BShap}}(\bar{x}, x', F) = \frac{1}{n!} \sum_{r \in \mathcal{R}} F(\bar{x} \backslash^{\{l | r_l \le i\}} x') - F(\bar{x} \backslash^{\{l | r_l \le (i-1)\}} x'). \tag{3}$$

## 3.3 Axioms

Eight axioms, which are known to be satisfied by IG (Lundstrom et al., 2022; Sundararajan et al., 2017), are defined below. We use the same names and definitions for the axioms as Lundstrom et al. (2022) and Lundstrom & Razaviyayn (2025), but we have slightly modified the wording for consistency. In all definitions, we use $A$ to denote any baseline attribution method that satisfies the axiom.

**Axiom 1** (Implementation Invariance)**:** $A$ is not a function of model implementation, but solely a function of the mathematical mapping of the model's domain to the range.

Implementation invariance is appealing because it ensures that networks that behave in the same way are also explained in the same way.

**Axiom 2** (Completeness)**:** If $(\bar{x}, x', F) \in \mathcal{D}_A$, then $\sum_{i=1}^n A_i(\bar{x}, x', F) = F(\bar{x}) - F(x')$.

Completeness provides an intuitive scale to the attribution values: The attribution value of a feature can be interpreted as how much that feature contributed to the change in neural network outcome.

**Axiom 3** (Sensitivity(a))**:** If $(\bar{x}, x', F) \in \mathcal{D}_A$, $F(\bar{x}) \ne F(x')$, and $\bar{x}, x'$ only vary in the $i$-th component, i.e. $\bar{x}_i \ne x'_i$, and $\bar{x}_j = x'_j \; \forall j \ne i$, then $A_i(\bar{x}, x', F) \ne 0$.

**Axiom 4** (Dummy/Sensitivity(b))**:** If $(\bar{x}, x', F) \in \mathcal{D}_A$ and $\partial_i F \equiv 0$, then $A_i(\bar{x}, x', F) = 0$.

Sensitivity(a) ensures that when only one feature is different between the baseline and the explicant, and this difference results in a difference in neural network outcome, that feature is not assigned a zero value in the attribution map. Dummy/Sensitivity(b) ensures that features that do not affect the neural network outcome are assigned a value of zero.

**Axiom 5** (Linearity): If $(\bar{x}, x', F), (\bar{x}, x', G) \in \mathcal{D}_A$ and $\alpha, \beta \in \mathbb{R}$, then $(\bar{x}, x', \alpha F + \beta G) \in \mathcal{D}_A$ and $A(\bar{x}, x', \alpha F + \beta G) = \alpha A(\bar{x}, x', F) + \beta A(\bar{x}, x', G)$.

**Axiom 6** (Symmetry-Preserving): Suppose that $(\bar{x}, x', F) \in \mathcal{D}_A$ and $i$ and $j$ are indices. Let $S_{ij}(x)$ be the function that swaps the values of $x_i$ and $x_j$. Then if $F(x) = F(S_{ij}(x))$ for any $x \in \mathcal{D}_F$, and $\bar{x} = S_{ij}(\bar{x})$ and $x' = S_{ij}(x')$, we have $A_i(\bar{x}, x', F) = A_j(\bar{x}, x', F)$.

Linearity ensures that if a neural network can be decomposed into a linear combination of subnetworks, the resulting attribution map can be decomposed in the same way. Symmetry-Preserving ensures that if two features are equal in the input and the baseline and are treated identically by the neural network, they should receive the same attribution value.

**Axiom 7** (Non-Decreasing Positivity): If $(\bar{x}, x', F) \in \mathcal{D}_A$ and $F$ is non-decreasing[1] from $x'$ to $\bar{x}$ then $A_i(\bar{x}, x', F) \geq 0$ for every index $i$.

Non-Decreasing Positivity avoids assigning negative attribution values when the neural network is non-decreasing from the baseline to the explicant.

**Axiom 8** (Affine Scale Invariance (ASI)): Suppose that $(\bar{x}, x', F) \in \mathcal{D}_A$, $\alpha, \beta \in \mathbb{R}$ with $\alpha \neq 0$, and $i$ is an index. Let $T$ be an affine transformation of element $i$, so that $T(x) := (x_1, \cdots, \alpha x_i + \beta, \cdots, x_n)$. Then we have $A(\bar{x}, x', F) = A(T(\bar{x}), T(x'), F \circ T^{-1})$.

Affine Scale Invariance (ASI) is appealing because it makes the attribution map invariant to affine unit changes; for example, when two networks perform equivalent computations but one expects an input in degrees Celsius and the other in degrees Fahrenheit, they will have the same attribution maps.

**Proposition 1** (IG satisfies Axioms 1-8 (Lundstrom et al., 2022; Sundararajan et al., 2017)). Let $A$ be the IG method as defined in Equation 2. Then it satisfies Axioms 1-8.

Below, we present the main theoretical results that MBS and BShap also satisfy these eight axioms.

**Proposition 2** (MBS satisfies Axioms 1-8). Let $A$ be the MBS method as defined in Equation 1. Then, for any $m \in R$ with $m > 0$, it satisfies Axioms 1-8.
*Proof.* Each axiom is proven separately in Lemmas 1, 2, 4, 6, 8, 10, 12, and 14 in Appendix B. □

**Proposition 3** (BShap satisfies Axioms 1-8). Let $A$ be the BShap method as defined in Equation 3. Then it satisfies Axioms 1-8.
*Proof.* Since Proposition 2 shows MBS satisfies Axioms 1-8 for any positive $m$, and MBS equals BShap when $m = 1$, it follows that BShap satisfies Axioms 1-8. □

BShap has already been proven to satisfy Axioms 2 (Completeness), 4 (Dummy/Sensitivity(b)), 5 (Linearity), 6 (Symmetry-Preserving), and 8 (Affine Scale Invariance) (Sundararajan & Najmi, 2020). Proposition 3 shows that BShap also satisfies Axioms 1 (Implementation invariance), 3 (Sensitivity(a)), and 7 (Non-Decreasing Positivity).

### 3.4 Monte Carlo Multi-Feature Baseline Shapley (MMBS)

The number of orderings in $\mathcal{R}$ grows factorially with the number of features, which rapidly grows to an infeasible amount to compute. For example, for the very small $28 \times 28$ pixel images of the MNIST (LeCun et al., 2010) dataset, the number of orderings is already approximately $3.2 \times 10^{1930}$.

To approximate the BShap method in a feasible amount of time, Yeh et al. (2022) proposed a Monte Carlo approach based on the ApproShapley method of Castro et al. (2009). We will call this method Monte Carlo BShap (MBShap). Instead of averaging over all possible orderings, it averages over a randomly sampled set

---

[1]A definition for non-decreasing based on (Lundstrom et al., 2022) is provided in Appendix B.7.

of orderings $\widetilde{\mathcal{R}}$:

$$A_i^{\text{MBShap}}(\bar{x}, x', F) = \frac{1}{|\widetilde{\mathcal{R}}|} \sum_{r \in \widetilde{\mathcal{R}}} F(\bar{x}^{\backslash\{l|r_l \leq i\}} x') - F(\bar{x}^{\backslash\{l|r_l \leq (i-1)\}} x'). \tag{4}$$

In the standard approach of calculating integrated gradients, the integral is approximated by a constant number of equally spaced steps (Sundararajan et al., 2017). We introduce an alternative approach using Monte Carlo estimation, which we call Monte Carlo Integrated Gradients (MIG). Here $\widetilde{\mathcal{Z}}$ is a set of randomly sampled values $\zeta$ from a uniform distribution over the range $[0, 1]$. The value of the integral is estimated by evaluating the gradient of $F$ for all values $\zeta$ in $\widetilde{\mathcal{Z}}$:

$$A_i^{\text{MIG}}(\bar{x}, x', F) = (\bar{x}_i - x'_i) \frac{1}{|\widetilde{\mathcal{Z}}|} \sum_{\zeta \in \widetilde{\mathcal{Z}}} \partial_i F((1 - \zeta)x' + \zeta\bar{x}). \tag{5}$$

By combining the ideas from MBShap and MIG, we can create a Monte Carlo estimator for MBS, which we call Monte Carlo Multi-Feature Baseline Shapley (MMBS). Here $\widetilde{\mathcal{S}}$ is a set of randomly sampled orderings $r$ and vectors $z$ of length $\lceil \frac{n}{m} \rceil$ with each element $z_k$ sampled from a uniform distributon over the range $[0, 1]$:

$$A_i^{\text{MMBS}}(\bar{x}, x', F) = \frac{1}{|\widetilde{\mathcal{S}}|} \sum_{(r,z) \in \widetilde{\mathcal{S}}} \sum_{k=1}^{\lceil \frac{n}{m} \rceil} ((\bar{x}^{\backslash\mathcal{I}(r,k)} x' - \bar{x}^{\backslash\mathcal{I}(r,k-1)} x')_i \tag{6}$$
$$\partial_i F((1 - z_k)(\bar{x}^{\backslash\mathcal{I}(r,k-1)} x') + z_k(\bar{x}^{\backslash\mathcal{I}(r,k)} x'))).$$

**Proposition 4.** MMBS is an unbiased estimator of MBS.

The proof for Proposition 4 is provided in Appendix A.

**Proposition 5** (MMBS satisfies some axioms)**.** Let $A$ be the MMBS method as defined in Equation 6. Then for any $m \in R$ with $m > 0$ and any $\widetilde{\mathcal{S}}$ it satisfies Axioms 1 (Implementation invariance), 4 (Dummy/Sensitivity(b)), 5 (Linearity), 7 (Non-Decreasing Positivity), and 8 (Affine Scale Invariance). However, there are values of $m \in R$ with $m > 0$ and $\widetilde{\mathcal{S}}$ for which it does not satisfy Axioms 2 (Completeness), 3 (Sensitivity(a)), and 6 (Symmetry Preserving).
    *Proof.* Each axiom is proven separately in Lemmas 1, 3, 5, 7, 9, 11, 13, and 15 in Appendix B.    □

Because of Proposition 5, MMBS always satisfies Axioms 1, 4, 5, 7, and 8, and because of Proposition 4, it satisfies all eight axioms in expectation.

### 3.5   Computational cost per sample

The definitions of MBShap, MIG, and MMBS in the previous section are all given for one feature. However, they are typically calculated for the entire image for every sample, which makes it possible to reuse neural network evaluations or gradient calculations. In MBShap, $n + 1$ evaluations of $F$ have to be calculated per sample. MIG only requires one gradient calculation per sample. In MMBS, the gradient of $F$ has to be calculated $\lceil \frac{n}{m} \rceil$ times per sample. Calculating a gradient is more computationally expensive than evaluating a neural network. However, when the step size $m$ is large, MMBS can be significantly faster than MBShap because of the lower number of calls.

### 3.6   The area under the deletion curve (AUDC) evaluation metric

The area under the deletion curve (AUDC) is a common metric to evaluate attribution maps (Petsiuk et al., 2018; Kapishnikov et al., 2019; 2021). A deletion curve shows how the outcome of the neural network changes when the features of $\bar{x}$ are replaced by corresponding features in an evaluation baseline $x'' \in \mathcal{D}_F$ in the order of their ranking in the attribution map. To calculate a deletion curve, first, the ranking $q$ of the features in the attribution map is determined: $q_i$ is 1 if feature $i$ is the feature with the highest attribution value, $q_i$ is 2 if feature $i$ has the second highest attribution, et cetera. The deletion curve $C_k(\bar{x}, x'', F, q)$ is defined as follows, where $k$ is the number of removed features:

$$C_k(\bar{x}, x'', F, q) = F(\bar{x}^{\backslash\{l|q_l \geq k\}} x''). \tag{7}$$

The area under the deletion curve (AUDC) is defined as follows:

$$M(\bar{x}, x'', F, q) = \frac{1}{n+1} \sum_{k=0}^{n} C_k(\bar{x}, x'', F, q).$$  (8)

When $F(\bar{x})$ is high, and $F(x'')$ is low, the AUDC can be used as a performance metric, and a lower AUDC is considered a better performance. The reasoning behind this is that higher attributed features in $\bar{x}$ should play a large role in the decision of the network. Setting these features to the value of the corresponding evaluation baseline feature should lower the outcome of the neural network more than changing a lower-ranked feature.

To lower the computation time, we approximated Equation 8 by sampling 200 equally spaced points along the deletion curve and assuming linear changes between these points. We implemented this using the `trapz` function in the NumPy Python library. In Appendix D, we show that increasing the number of deletion curve steps beyond 200 has little effect on the calculated AUDC scores.

## 4 Experiments

In this section, we will first describe the datasets and neural networks that were used (Section 4.1). Then, we will show how the step size and number of samples affect the outcome of MMBS, and that MMBS can be used to approximate BShap (Section 4.2). After that, we will compare AUDC scores of MMBS and several competing methods on three neural networks (Section 4.4). Finally, we will investigate the influence of the attribution and evaluation baselines on attribution maps and their AUDC scores (Section 4.5).

### 4.1 Datasets and neural networks

Evaluating an attribution method requires a dataset and a network trained on this dataset. We used the Fashion MNIST (Xiao et al., 2017) and ImageNet1k (Russakovsky et al., 2015) datasets. To enable fast prototyping on the Fashion MNIST dataset, we used a very lightweight convolutional neural network (CNN) architecture consisting of two convolutional layers and two fully connected layers (Oikarinen, 2021). On ImageNet1k, we used two neural network architectures: A ResNet with a depth of 50 layers (ResNet50) (He et al., 2016), and a Vision Transformer(ViT) with a patch size of 16x16 pixels (ViT-B/16) (Dosovitskiy et al., 2021). For both networks on ImageNet1K, we used pretrained weights from the torchvision library (Marcel & Rodriguez, 2010). On the grayscale images of Fashion MNIST, each pixel is a feature. In the color images of ImageNet1K, each color subpixel is a feature. When displaying the attribution maps of color images, we show the sum of the different color subpixels.

### 4.2 MMBS as an approximation of BShap

To measure the convergence of MMBS to BShap we introduce the Relative Mean Squared Error(MSE) between a heatmap $y \in \mathbb{R}^n$ and a reference heatmap $\hat{y} \in \mathbb{R}^n$:

$$\frac{\|y - \hat{y}\|^2}{\|\hat{y}\|^2}$$  (9)

The relative MSE is 1 for a heatmap $y$ of all zeros, and 0 when $y = \hat{y}$. To determine whether MMBS converges to BShap, the ideal approach would be to use the BShap heatmap as a reference ($\hat{y}$). However, this is impractical for all three networks due to BShap's prohibitively long computation time.

For the Fashion MNIST network, we approximated BShap by running 10,000 iterations of MBShap, and used this as a reference $\hat{y}$. This approximation was computed for 40 randomly sampled images from the dataset. For each of these images, the Relative MSE was evaluated at each iteration for MMBS across various step sizes (1, 2, 4, 8, 16, 32, 64, and 128) and for an independent run of MBShap. When we say that MMBS was calculated with $k$ steps for $l$ iterations, we mean that the step size was set to $m = \frac{n}{k}$ so that $\lceil \frac{n}{m} \rceil = k$, and that $l$ orderings were sampled ($|\widetilde{\mathcal{S}}| = l$). The results are presented in Figure 2. The relative MSE between the two MBShap attribution maps at 10,000 iterations is close to zero, suggesting that MBShap is almost fully converged at 10,000 iterations for the Fashion MNIST network. The relative MSE of MMBS with 8 or

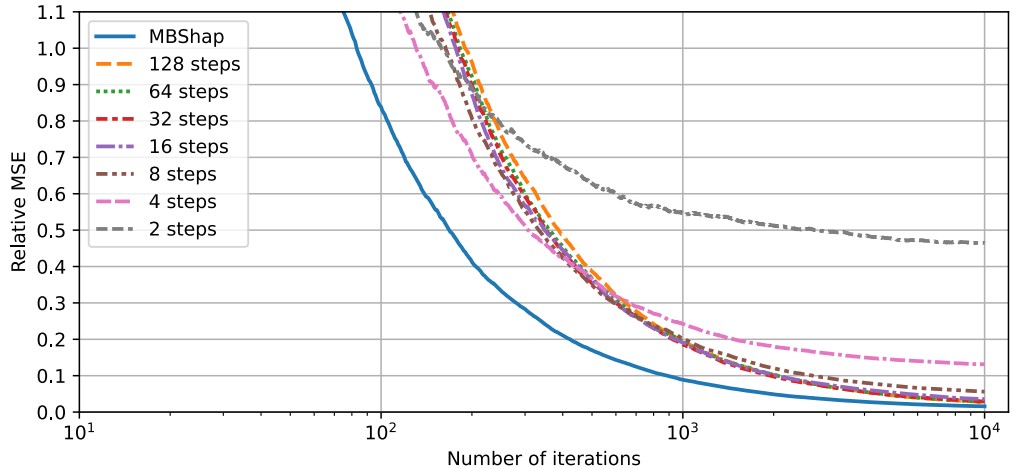

Figure 2: This figure shows how the relative MSE of MMBS at different numbers of steps and MBShap changes with the number of iterations. The plotted relative MSE metric is the average over 40 images of the Fashion MNIST dataset, and it used 10.000 iterations of MBShap as a reference. The MBShap curve was calculated independently from the reference. MMBS with 1 step had a Relative MSE that was consistently higher than 1.1, so it was not included in the plot.

more steps at 10,000 iterations is also very close to zero, indicating that MMBS can closely approximate BShap.

Figure 3 shows examples of attribution maps on the Fashion MNIST dataset at different numbers of steps and iterations. It shows that even when MMBS is not fully converged, it can already be visually similar to MBShap.

Calculating 10.000 iterations of MBShap was not possible for the ImageNet networks because the larger images and the more complex networks would make the computation time too long. Therefore, it was not possible to compare MMBS to an almost fully converged MBShap result on these networks. Appendix C includes convergence plots using different references: 100 or 30 iterations of BShap, and 10.000 iterations of MMBS with 128 steps. Those results suggest that MMBS exhibits similar convergence behavior on ImageNet as on Fashion MNIST.

### 4.3 Runtime

The runtime of MBShap and of MMBS with different numbers of steps (1, 2, 4, 8, 16, 32, 64, and 128) was measured by calculating one iteration on 50 different images on each dataset. The used computational hardware was an Nvidia Titan RTX GPU and an Intel Xeon Gold 6130 CPU with 16 cores running at 2.10GHz. The results are shown in Table 1. The longer runtime of the methods on the ViT compared to the ResNet can be explained by two factors: (1) the ViT is a larger network than the ResNet (86.9M vs 25.6M parameters), and (2) the ResNet cropped the input images to $224 \times 224$, while the ViT cropped the input images to $384 \times 384$, so each ordering contained almost three times as many features for the ViT.

### 4.4 Comparison with other methods

We compared MMBS to the existing methods of IG (Sundararajan et al., 2017), GIG (Kapishnikov et al., 2021), XRAI (Kapishnikov et al., 2019), and Grad-CAM (Selvaraju et al., 2017). MMBS, IG, and GIG were also paired with SmoothGrad (Smilkov et al., 2017). Additionally, randomly sampled orderings were used as attribution maps, to check if the methods performed better than random.

For MMBS, we used 8 steps, 1024 samples, and an all-zero baseline. For IG, we used 256 steps and an all-zero baseline. For GIG, we used the implementation from the Saliency library, which is developed by the same group that wrote the GIG paper, but it uses a slightly different approach to bound the maximum step size, and by default, it uses a higher fraction of features that can be modified in each step (25% instead of 10%). We calculated the unbounded version of GIG as described in the paper (GIG (paper)), and the default

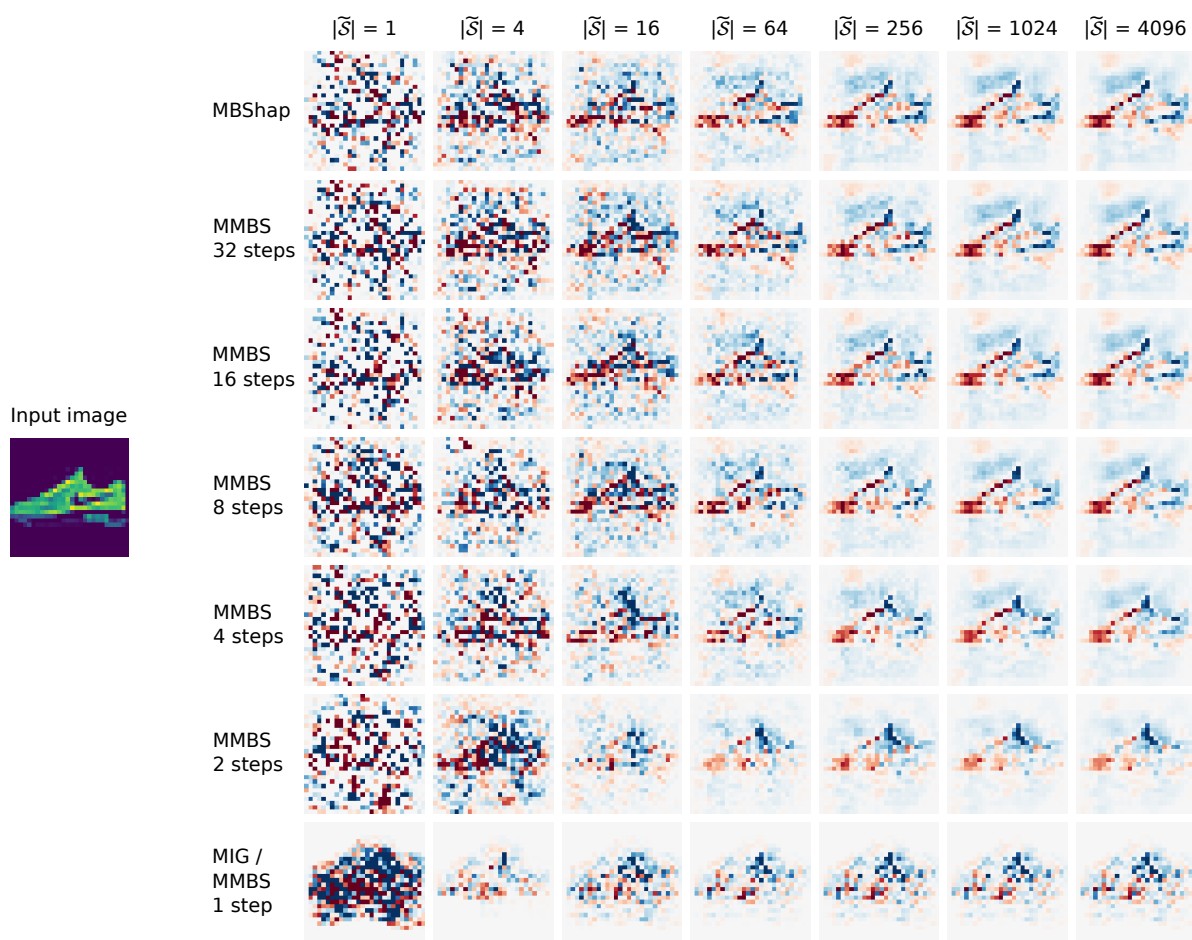

Figure 3: This figure shows heatmaps of an image of a sneaker calculated using the MIG, MBShap, and MMBS methods for a small CNN trained on Fashion MNIST. The same orderings $r$ were used when calculating the MBShap and MMBS attribution maps. All images use the same colormap where blue denotes positive attributions and red denotes negative attributions.

Table 1: Comparison of calculation time in seconds of one iteration of MMBS and MBShap on different datasets. The table lists the mean and the standard deviation over 50 images.

| Method | Fashion MNIST runtime (s) | ImageNet ResNet runtime (s) | ImageNet ViT runtime (s) |
|---|---|---|---|
| MBShap | $0.3330 \pm 0.0395$ | $695.60 \pm 11.230$ | $6064.4 \pm 6.9381$ |
| MMBS (1 step) | $0.0021 \pm 0.0003$ | $0.0206 \pm 0.0035$ | $0.0446 \pm 0.0073$ |
| MMBS (2 steps) | $0.0035 \pm 0.0009$ | $0.0434 \pm 0.0096$ | $0.0732 \pm 0.0067$ |
| MMBS (4 steps) | $0.0064 \pm 0.0016$ | $0.1108 \pm 0.0230$ | $0.1368 \pm 0.0039$ |
| MMBS (8 steps) | $0.0138 \pm 0.0035$ | $0.2281 \pm 0.0466$ | $0.2794 \pm 0.0045$ |
| MMBS (16 steps) | $0.0396 \pm 0.0089$ | $0.4591 \pm 0.0902$ | $0.5597 \pm 0.0113$ |
| MMBS (32 steps) | $0.0954 \pm 0.0218$ | $0.9262 \pm 0.1695$ | $1.1211 \pm 0.0163$ |
| MMBS (64 steps) | $0.1968 \pm 0.0430$ | $1.8527 \pm 0.3381$ | $2.2429 \pm 0.0296$ |
| MMBS (128 steps) | $0.3954 \pm 0.0836$ | $3.7013 \pm 0.6759$ | $4.4809 \pm 0.0592$ |

values of the Saliency library (GIG (Saliency)). In both versions, we used 256 steps and an all-zero baseline. For SmoothGrad, we used a noise standard deviation of $0.15(\max(\bar{x}) - \min(\bar{x}))$, and 25 samples. For MMBS + SG, we used 128 instead of 1024 samples in each MMBS calculation to reduce the total computation time.

Table 2: Comparison of AUDC scores on different datasets. The table lists the mean and the 5th and 95th percentiles. The best mean score for each network is shown in boldface.

| Method | Fashion MNIST AUDC | ImageNet ResNet AUDC | ImageNet ViT AUDC |
|---|---|---|---|
| MMBS (ours) | **0.057** [0.001, 0.184] | **0.014** [0.001, 0.055] | **0.023** [0.000, 0.096] |
| MMBS + SG | 0.060 [0.002, 0.181] | 0.023 [0.001, 0.084] | 0.028 [0.000, 0.108] |
| IG | 0.216 [0.003, 0.785] | 0.060 [0.001, 0.218] | 0.118 [0.001, 0.468] |
| IG + SG | 0.093 [0.003, 0.264] | 0.033 [0.001, 0.151] | 0.075 [0.001, 0.399] |
| GIG (paper) | 0.341 [0.003, 0.897] | 0.102 [0.001, 0.388] | 0.071 [0.001, 0.466] |
| GIG (paper) + SG | 0.161 [0.003, 0.654] | 0.032 [0.001, 0.156] | 0.024 [0.000, 0.066] |
| GIG (Saliency) | 0.207 [0.003, 0.783] | 0.044 [0.001, 0.203] | 0.106 [0.001, 0.430] |
| GIG (Saliency) + SG | 0.088 [0.002, 0.250] | 0.029 [0.001, 0.119] | 0.067 [0.001, 0.363] |
| XRAI (B + W) | 0.377 [0.035, 0.914] | 0.158 [0.004, 0.417] | 0.388 [0.015, 0.822] |
| XRAI (zero) | 0.301 [0.021, 0.844] | 0.166 [0.005, 0.422] | 0.379 [0.016, 0.806] |
| GradCAM | 0.520 [0.147, 0.962] | 0.124 [0.004, 0.354] | N.A |
| Random | 0.537 [0.113, 0.877] | 0.160 [0.004, 0.389] | 0.514 [0.025, 0.848] |

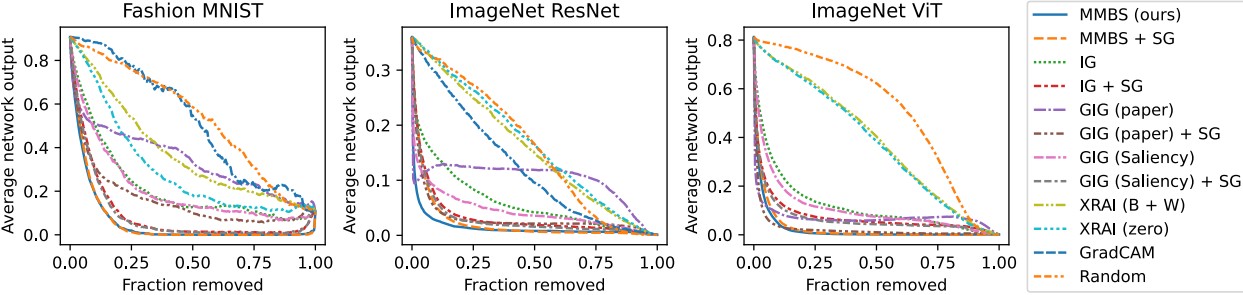

Figure 4: The average deletion curves for different attribution map methods on three networks.

For XRAI, we tried two variants: XRAI (B + W) is based on the average of IG with a black baseline and IG with a white baseline, which is how XRAI was used in its original publication (Kapishnikov et al., 2019). XRAI (zero) is based on IG with an all-zero baseline. Grad-CAM was not applied to the ViT model because it did not have a suitable CNN architecture.

Each method was applied to all three neural networks, and AUDC scores were calculated using a baseline of all zeros. From the Fashion MNIST dataset, 1000 randomly sampled images from the test set were used. From the ImageNet dataset, the first image from each class in the validation set was used, to ensure that images from all classes were included, also resulting in 1000 images. To avoid including increasing deletion curves, AUDC scores were only included when the network outcome on the input image was higher than on the evaluation baseline, which was the case for 986, 999, and 993 images, respectively, on the Fashion MNIST, ResNet, and ViT networks.

The average deletion curves are shown in Figure 4 and the AUDC scores are shown in Table 2. MMBS had the lowest AUDC on all three networks. GIG (paper) + SmoothGrad had a very similar mean, and a slightly lower 95th percentile on the ViT network, but it scored worse than MMBS on the other two networks. Another interesting observation is that IG and both versions of GIG had a lower AUDC on all networks when combined with SmoothGrad. However, when MMBS was combined with SmoothGrad, the AUDC scores became slightly higher.

### 4.5 The effect of the baselines on the attribution and deletion curve

Like IG (Sturmfels et al., 2020), the result of MMBS depends on the chosen baseline. To illustrate this, we calculated IG and MMBS attribution maps of the ResNet50 network for five different baselines on an image of a toucan, which has a black and an almost white region. The baselines are: zeros before normalization

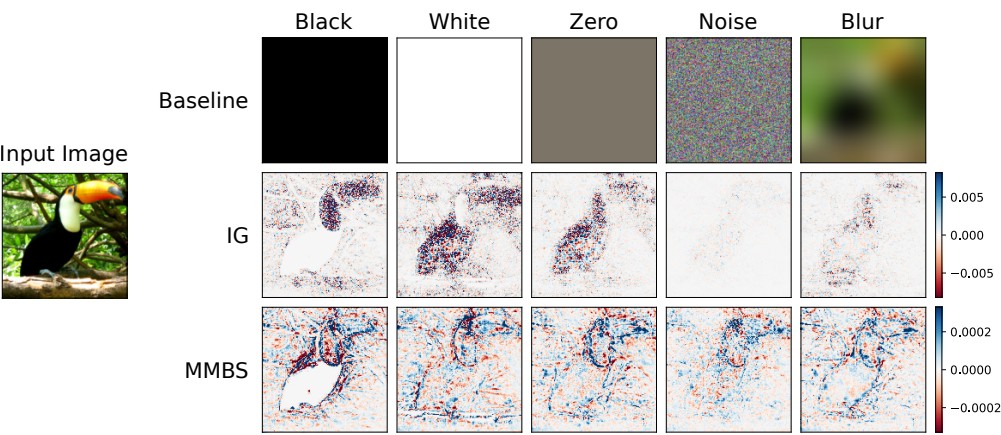

Figure 5: IG and MMBS Attribution map results with different baselines. MMBS was configured with 8 steps and 1024 samples, and IG with 256 steps. With both methods, the outcome is affected by the baseline.

Table 3: Area under deletion curve (AUDC) metrics for different combinations of baselines used in the calculation (rows) and evaluation (columns) of IG and MMBS attribution maps. The table lists the mean and the 5th and 95th percentiles. The best mean score for each AUDC metric is shown in bold numbers.

| Base-line | Method | Black AUDC | White AUDC | Zero AUDC | Noise AUDC | Blur AUDC |
|---|---|---|---|---|---|---|
| Black | IG | 0.031 [0.001, 0.105] | 0.080 [0.004, 0.226] | 0.114 [0.004, 0.313] | 0.063 [0.002, 0.191] | 0.146 [0.004, 0.390] |
|  | MMBS | **0.011** [0.001, 0.040] | 0.183 [0.004, 0.517] | 0.128 [0.002, 0.415] | 0.055 [0.001, 0.228] | 0.160 [0.002, 0.460] |
| White | IG | 0.089 [0.004, 0.243] | 0.028 [0.001, 0.093] | 0.103 [0.002, 0.301] | 0.056 [0.002, 0.179] | 0.131 [0.003, 0.355] |
|  | MMBS | 0.172 [0.005, 0.482] | **0.011** [0.001, 0.041] | 0.097 [0.001, 0.359] | 0.027 [0.001, 0.116] | 0.120 [0.002, 0.390] |
| Zero | IG | 0.077 [0.003, 0.219] | 0.057 [0.002, 0.176] | 0.060 [0.001, 0.220] | 0.045 [0.001, 0.152] | 0.092 [0.002, 0.303] |
|  | MMBS | 0.090 [0.002, 0.311] | 0.070 [0.002, 0.274] | **0.014** [0.001, 0.052] | 0.013 [0.001, 0.047] | 0.035 [0.001, 0.179] |
| Noise | IG | 0.050 [0.001, 0.183] | 0.028 [0.001, 0.104] | 0.049 [0.001, 0.212] | 0.009 [0.001, 0.029] | 0.076 [0.001, 0.278] |
|  | MMBS | 0.046 [0.001, 0.204] | 0.020 [0.001, 0.081] | 0.018 [0.001, 0.069] | **0.005** [0.000, 0.015] | 0.037 [0.001, 0.173] |
| Blur | IG | 0.057 [0.002, 0.182] | 0.043 [0.002, 0.135] | 0.068 [0.002, 0.249] | 0.039 [0.001, 0.122] | 0.071 [0.001, 0.303] |
|  | MMBS | 0.074 [0.002, 0.257] | 0.060 [0.002, 0.217] | 0.027 [0.001, 0.128] | 0.017 [0.001, 0.068] | **0.016** [0.001, 0.080] |

(Black), ones before normalization (White), zeros after normalization (Zero), uniform noise over the full range of possible colors (Noise), and a Gaussian blurred version of the input image with a $\sigma$ of 25 pixels (Blur). Neural networks are often normalized so that the inputs of the training set have a mean of zero, which was also done in this paper. Therefore, a black image does not correspond to an input value of all zeros. The results are shown in Figure 5. When applying the neural network on the baseline, the output probability on the class toucan was less than 0.5% on all baselines. Nevertheless, the attribution maps look different for each baseline.

When evaluating attribution methods using the AUDC, there are two baselines: the attribution baseline $x'$, and the evaluation baseline $x''$. We calculated AUDC values for IG and MMBS using the ResNet50 network for every combination of the five baseline choices from the previous paragraph (and Figure 5). We used the first image from each class in the validation set of ImageNet1k. To avoid including increasing deletion curves, AUDC scores were only included when the network outcome on the input image was higher than on the evaluation baseline, which was the case for 998, 999, 999, 999, and 998 images for the black, white, zero, noise, and blur evaluation baselines, respectively. Table 3 shows the results of this experiment. For each evaluation baseline, the lowest AUDC was achieved by MMBS using the same attribution baseline.

## 5 Discussion and future work

In this section, we will first discuss how the differences between IG and BShap may be characterized by their different mathematical axioms (Section 5.1). We will then briefly discuss the potential of MMBS in other applications than image classification (Section 5.2). After that, we will discuss how to choose a baseline, and whether distributions of baselines may be a better choice for modeling neutrality (Section 5.3).

## 5.1 Additional axioms of IG and BShap

In Section 3.3, we proved that IG, BShap, and MMBS have several axioms in common. However, the experiments in Section 4 make it apparent that IG and MMBS/BShap can provide significantly different attributions. This implies that IG and BShap each satisfy additional axioms that the other method does not.

Lundstrom & Razaviyayn (2025) proved several additional axioms of IG. One way to distinguish a method from all other methods is to prove that it is the only method that satisfies a certain combination of axioms. This is called a unique characterization. Lundstrom & Razaviyayn (2025) also presented four ways to uniquely characterize IG. For many of the axioms used in these unique characterizations, we know that MBS and BShap satisfy them too, so MBS and BShap have to not satisfy at least one axiom of the remaining axioms in each characterization. Using this reasoning, we can derive that MBS and BShap do not satisfy the axioms of Proportionality and Symmetric Monotonicity.

In the cost-sharing literature, several additional axioms of the Shapley-Shubik (SS) method (Shubik, 1962) have been studied. SS is equal to BShap when you assume that the baseline $x'$ consists of all zeros, that $F(x') = 0$, and that $F$ is non-decreasing, positive, and continuously differentiable. Friedman & Moulin (1999) showed that SS has the axiom of demand monotonicity, and the Aumann-Shapley method (and therefore also IG) does not have this axiom. Moreover, they provided a unique characterization of SS using this axiom. Sprumont (1998) showed that SS satisfies the axiom of Ordinality, which is a generalization of the Affine Scale Invariance axiom. They also provided a unique characterization of SS using this axiom. It would be interesting future work to investigate whether these results can be generalized to BShap on the broader function space $\mathcal{F}^2$ used in this paper and by Lundstrom & Razaviyayn (2025). Sundararajan & Najmi (2020) showed that BShap maintains the axioms of SS when the baseline is not zero, but they did not address that neural networks are often not nondecreasing, positive, and continuously differentiable.

## 5.2 Other applications than image classification

In this paper, we focused on experiments with image classification problems. This was done to limit the scope of the paper and because image classification is a very popular problem to explain using attribution methods. However, the MMBS method can also be used in the attribution of other models. It would be interesting future work to investigate whether MMBS is also a close approximation of MBShap for other types of inputs, such as the problems surveyed by Chen et al. (2023).

## 5.3 Defining a neutral baseline

Similar to much of the existing literature (Sundararajan et al., 2017; Sundararajan & Najmi, 2020; Kapishnikov et al., 2021), we assume in this paper that a single image can serve as a neutral baseline. This is the case in the original cost-sharing problem, where the outcome of $F$ is guaranteed to be zero on an all-zero input (Shubik, 1962). Moreover, in certain cases, a specific baseline may be used to obtain a specific explanation of a neural network (Mamalakis et al., 2023), or to explain only the changes over time (Schut et al., 2026). However, in many cases, one baseline does not accurately capture the concept of neutrality (Sturmfels et al., 2020). In image classification models, the outputs always sum to one, so it's impossible to have a baseline that has a value of zero for all classes. In models trained on ImageNet1k, this is alleviated by the large number of classes, so any baseline is likely to have a network output close to zero for almost all classes. Still, in Section 4.5, we showed that different baselines may lead to strongly different attribution maps even when, for all these baselines, the network output on the target class is low.

An alternative definition of a neutral baseline is the data distribution conditioned on the features that haven't been removed (Lundberg & Lee, 2017). BShap, MMBS, and deletion curves could be adapted to this definition of neutrality by averaging in every step over many random samples from the conditional data distribution. In practice, the conditional data distribution is often unavailable, but it may be interesting future work to approximate it by conditional sampling using a diffusion model (Lugmayr et al., 2022) or another generative AI model. A downside of using conditional baselines is that it may break Axiom 4 (Dummy/Sensitivity(b)), because a feature that is unused by the network may affect the baseline value of features that are used by the network, resulting in a non-zero attribution (Janzing et al., 2020). Some existing approaches that are used in the calculation of attribution maps or deletion curves can be considered to approximate the conditional

data distribution: The Expected Gradients method (Erion et al., 2021) averaged over IG attribution maps with baselines that were randomly sampled from the training set, which closely approximates sampling from the unconditional data distribution. Rong et al. (2022) used inpainting of the missing pixels to calculate deletion curves, so the results are conditioned on the data, but not random. MMBS showed a good AUDC performance (Table 3) when the attribution and evaluation baselines were the same. It would be interesting future work to investigate whether AUDC scores are also low when neutrality is modeled in other ways, as long as it is the same in the calculation of the attribution map and the AUDC.

## 6    Conclusion

In this paper, we presented the MBS and MMBS methods for calculating attribution maps. MBS generalizes the IG and BShap methods and satisfies eight axioms that IG and BShap also satisfy. MMBS is an unbiased estimator of MBS. On image classification tasks, we showed that MMBS can not only serve as a fast approximation to MBS but also to BShap, yielding results similar to MBShap while being orders of magnitude faster to compute. Moreover, we compared MMBS with existing methods across three image classification networks, and it achieved the lowest AUDC metric on all three networks. All in all, MMBS is an attribution method with a strong theoretical foundation and an acceptable computation time that yields state-of-the-art AUDC scores.

### Acknowledgements

This work was funded by the Dutch Research Council (NWO) through the UTOPIA project (EN-WSS.2018.003). We gratefully acknowledge Lasse Veenstra for his comments on improving the proofs and Lisa Meijer for her comments on improving the writing.

### Code and data availability

All code used in the experiments of this paper is included in the supplementary material and on GitHub (https://github.com/D1rk123/MMBS). The ImageNet1k (Russakovsky et al., 2015) and Fashion MNIST (Xiao et al., 2017) datasets are already publicly available.

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

## A    Proof that MMBS is an unbiased estimator of MBS

We will first prove that MMBS with one sample ($\widetilde{\mathcal{S}} = \{(\tilde{r}, \tilde{z})\}$) is an unbiased estimator of MBS. $\tilde{r}$ and every element $\tilde{z}_k$ of $\tilde{z}$ are independent random variables, so the expected values can be split. $\tilde{r}$ is sampled from a distribution that has a probability of $\frac{1}{n!}$ for each ordering in $\mathcal{R}$. Each $\tilde{z}_k$ is sampled from a uniform distribution over the range $[0, 1]$.

$$
\mathbb{E}_{\tilde{r}, \tilde{z}} \left[ \sum_{k=1}^{\lceil \frac{n}{m} \rceil} ((\bar{x}\backslash^{\mathcal{I}(\tilde{r},k)} x' - \bar{x}\backslash^{\mathcal{I}(\tilde{r},k-1)} x')_i \partial_i F((1 - \tilde{z}_k)(\bar{x}\backslash^{\mathcal{I}(\tilde{r},k-1)} x') + \tilde{z}_k(\bar{x}\backslash^{\mathcal{I}(\tilde{r},k)} x'))) \right]
$$

$$
= \mathbb{E}_{\tilde{r}} \left[ \sum_{k=1}^{\lceil \frac{n}{m} \rceil} (\bar{x}\backslash^{\mathcal{I}(\tilde{r},k)} x' - \bar{x}\backslash^{\mathcal{I}(\tilde{r},k-1)} x')_i \, \mathbb{E}_{\tilde{z}_k} \left[ \partial_i F((1 - \tilde{z}_k)(\bar{x}\backslash^{\mathcal{I}(\tilde{r},k-1)} x') + \tilde{z}_k(\bar{x}\backslash^{\mathcal{I}(\tilde{r},k)} x')) \right] \right]
$$

$$
= \mathbb{E}_{\tilde{r}} \left[ \sum_{k=1}^{\lceil \frac{n}{m} \rceil} (\bar{x}\backslash^{\mathcal{I}(\tilde{r},k)} x' - \bar{x}\backslash^{\mathcal{I}(\tilde{r},k-1)} x')_i \int_{\zeta=0}^{1} \partial_i F((1 - \zeta)(\bar{x}\backslash^{\mathcal{I}(\tilde{r},k-1)} x') + \zeta(\bar{x}\backslash^{\mathcal{I}(\tilde{r},k)} x')) d\zeta \right] \tag{10}
$$

$$
= \mathbb{E}_{\tilde{r}} \left[ \sum_{k=1}^{\lceil \frac{n}{m} \rceil} A^{\mathrm{IG}}(\bar{x}\backslash^{\mathcal{I}(\tilde{r},k)} x', \bar{x}\backslash^{\mathcal{I}(\tilde{r},k-1)} x', F) \right]
$$

$$
= \frac{1}{n!} \sum_{r \in \mathcal{R}} \sum_{k=1}^{\lceil \frac{n}{m} \rceil} A^{\mathrm{IG}}(\bar{x}\backslash^{\mathcal{I}(r,k)} x', \bar{x}\backslash^{\mathcal{I}(r,k-1)} x', F)
$$

$$
= A_i^{\mathrm{MBS}}(\bar{x}, x', F)
$$

MMBS with multiple samples is the average of multiple independent single-sample MMBS calls. Since these are all unbiased estimators, their average is also unbiased. □

## B    Proofs that MBS and MMBS satisfy axioms

### B.1    Implementation invariance

**Definition of Implementation Invariance:** $A$ is not a function of model implementation, but solely a function of the mathematical mapping of the model's domain to the range.

**Lemma 1.** MBS and MMBS satisfy implementation invariance.
*Proof.* MBS and MMBS only use the output or the gradient of the full model ($F$) in their calculations. Therefore, they satisfy implementation invariance. □

### B.2    Completeness

**Definition of Completeness:** If $(\bar{x}, x', F) \in \mathcal{D}_A$, then $\sum_{i=1}^{n} A_i(\bar{x}, x', F) = F(\bar{x}) - F(x')$.

**Lemma 2.** MBS satisfies completeness.
*Proof.* Because IG satisfies completeness (Lundstrom & Razaviyayn, 2025), every IG call within MBS can be

written as the difference between two calls to $F$. By summing all IG calls with the same ordering $r$, most of these calls will cancel out, except the first startpoint $F(x')$ and the last endpoint $F(\bar{x})$. For every $r \in \mathcal{R}$, this sum satisfies completeness, so MBS satisfies completeness.

$$
\begin{aligned}
\sum_{i=1}^{n} A_i^{\text{MBS}}(\bar{x}, x', F) &= \sum_{i=1}^{n} \frac{1}{n!} \sum_{r \in \mathcal{R}} \sum_{k=1}^{\lceil \frac{n}{m} \rceil} A_i^{\text{IG}}(\bar{x} \backslash^{\mathcal{I}(r,k)} x', \bar{x} \backslash^{\mathcal{I}(r,k-1)} x', F) \\
&= \frac{1}{n!} \sum_{r \in \mathcal{R}} \sum_{k=1}^{\lceil \frac{n}{m} \rceil} \sum_{i=1}^{n} A_i^{\text{IG}}(\bar{x} \backslash^{\mathcal{I}(r,k)} x', \bar{x} \backslash^{\mathcal{I}(r,k-1)} x', F) \\
&= \frac{1}{n!} \sum_{r \in \mathcal{R}} \sum_{k=1}^{\lceil \frac{n}{m} \rceil} F(\bar{x} \backslash^{\mathcal{I}(r,k)} x') - F(\bar{x} \backslash^{\mathcal{I}(r,k-1)} x') \\
&= \frac{1}{n!} \sum_{r \in \mathcal{R}} F(\bar{x}) - F(x') \\
&= F(\bar{x}) - F(x')
\end{aligned}
\tag{11}
$$

$\square$

**Lemma 3.** MMBS does not always satisfy completeness.

*Proof.* A counteraxample can be constructed using a network with single input variable $x_1$ consisting of a single rectified linear unit $F = \text{ReLU}(x_1)$, $\bar{x}_1 = 1$, $x'_1 = -2$ when calculating MMBS with one sample ($|\widetilde{\mathcal{S}}| = 1$), where $r = (1)$ and $z = \frac{1}{2}$. In that case:

$$
\begin{aligned}
A_1^{\text{MMBS}}(\bar{x}, x', F) &= \frac{1}{|\widetilde{\mathcal{S}}|} \sum_{(r,z) \in \widetilde{\mathcal{S}}} \sum_{k=1}^{\lceil \frac{n}{m} \rceil} ((\bar{x} \backslash^{\mathcal{I}(r,k)} x' - \bar{x} \backslash^{\mathcal{I}(r,k-1)} x')_1 \\
&\qquad\qquad\qquad \partial_1 F((1 - z_k)(\bar{x} \backslash^{\mathcal{I}(r,k-1)} x') + z_k(\bar{x} \backslash^{\mathcal{I}(r,k)} x'))) \\
&= (1 - (-2)) \cdot \partial_1 F(\frac{1}{2}(-2) + \frac{1}{2}(1)) \\
&= 3 \cdot \partial_1 F(-\frac{1}{2}) = 3 \cdot 0 = 0.
\end{aligned}
\tag{12}
$$

This is unequal to $F(\bar{x}) - F(x') = 1 - 0 = 1$. Therefore, MMBS does not satisfy completeness. $\square$

### B.3 Sensitivity(a)

**Definition of Sensitivity(a):** If $(\bar{x}, x', F) \in \mathcal{D}_A$, $F(\bar{x}) \neq F(x')$, and $\bar{x}, x'$ only vary in the $i$-th component, i.e. $\bar{x}_i \neq x'_i$, and $\bar{x}_j = x'_j \ \forall j \neq i$, then $A_i(\bar{x}, x', F) \neq 0$.

**Lemma 4.** MBS satisfies Sensitivity(a).

*Proof.* The value of $(\bar{x} \backslash^{\mathcal{I}(r,k)} x')_j$ is always equal to either $\bar{x}_j$ or $x'_j$. Therefore, if $\bar{x}_j = x'_j$, then $(\bar{x} \backslash^{\mathcal{I}(r,k)} x')_j = \bar{x}_j = x'_j$ for all values of $k$, which in turn causes $A_j^{\text{MBS}}(\bar{x}, x', F)$ to be zero:

$$
\begin{aligned}
&A_j^{\text{MBS}}(\bar{x}, x', F) \\
&= \frac{1}{n!} \sum_{r \in \mathcal{R}} \sum_{k=1}^{\lceil \frac{n}{m} \rceil} A_j^{\text{IG}}(\bar{x} \backslash^{\mathcal{I}(r,k)} x', \bar{x} \backslash^{\mathcal{I}(r,k-1)} x', F) \\
&= \frac{1}{n!} \sum_{r \in \mathcal{R}} \sum_{k=1}^{\lceil \frac{n}{m} \rceil} ((\bar{x} \backslash^{\mathcal{I}(r,k)} x')_j - (\bar{x} \backslash^{\mathcal{I}(r,k-1)} x')_j) \int_{\zeta=0}^{1} \partial_j F((1 - \zeta)(\bar{x} \backslash^{\mathcal{I}(r,k-1)} x') + \zeta(\bar{x} \backslash^{\mathcal{I}(r,k)} x')) d\zeta \\
&= \frac{1}{n!} \sum_{r \in \mathcal{R}} \sum_{k=1}^{\lceil \frac{n}{m} \rceil} (\bar{x}_j - \bar{x}_j) \int_{\zeta=0}^{1} \partial_j F((1 - \zeta)(\bar{x} \backslash^{\mathcal{I}(r,k-1)} x') + \zeta(\bar{x} \backslash^{\mathcal{I}(r,k)} x')) d\zeta \\
&= 0.
\end{aligned}
\tag{13}
$$

If $\bar{x}_j = x'_j$ for all values of $j$ except $j = i$, then $A_i^{\mathrm{MBS}}(\bar{x}, x', F)$ is the only value that may not be zero. In that case, because MBS satisfies the completeness axiom, $A_i^{\mathrm{MBS}}(\bar{x}, x', F)$ has to be equal to $F(\bar{x}) - F(x')$, so, when also $F(\bar{x}) \neq F(x')$, then $A_i^{\mathrm{MBS}}(\bar{x}, x', F) \neq 0$ □

**Lemma 5.** MMBS does not always satisfy Sensitivity(a).

*Proof.* The same counterexample can be used as for Completeness (Lemma 3). $\bar{x}_i \neq x'_i$ for only one value of $i$ because the only possible value of $i$ is 1. Moreover, $F(\bar{x}) = 1$ and $F(x') = 0$, so $F(\bar{x}) \neq F(x')$. Yet, $A_1^{\mathrm{MMBS}}(\bar{x}, x', F) = 0$. Therefore, MMBS does not satisfy Sensitivity(a). □

## B.4 Dummy/Sensitivity(b)

**Definition of Dummy/Sensitivity(b):** If $(\bar{x}, x', F) \in \mathcal{D}_A$ and $\partial_i F \equiv 0$, then $A_i(\bar{x}, x', F) = 0$.

**Lemma 6.** MBS satisfies Dummy/Sensitivity(b).

*Proof.* The $i$-th element of MBS is only calculated from the $i$-th components of the IG steps. Therefore, the dummy axiom of MBS can be proven from the fact that IG satisfies dummy (Lundstrom & Razaviyayn, 2025). When $\partial_i F \equiv 0$ then:

$$
\begin{aligned}
A_i^{\mathrm{MBS}}(\bar{x}, x', F) &= \frac{1}{n!} \sum_{r \in \mathcal{R}} \sum_{k=1}^{\lceil \frac{n}{m} \rceil} A_i^{\mathrm{IG}}(\bar{x}\backslash^{\mathcal{I}(r,k)} x', \bar{x}\backslash^{\mathcal{I}(r,k-1)} x', F) \\
&= \frac{1}{n!} \sum_{r \in \mathcal{R}} \sum_{k=1}^{\lceil \frac{n}{m} \rceil} 0 \\
&= 0.
\end{aligned}
\tag{14}
$$

□

**Lemma 7.** MMBS satisfies Dummy/Sensitivity(b).

*Proof.* The $i$-th element of MMBS is only calculated from terms that are multiplied with $\partial_i F$. Therefore, when $\partial_i F \equiv 0$ then:

$$
\begin{aligned}
A_i^{\mathrm{MMBS}}(\bar{x}, x', F) &= \frac{1}{|\widetilde{\mathcal{S}}|} \sum_{(r,z) \in \widetilde{\mathcal{S}}} \sum_{k=1}^{\lceil \frac{n}{m} \rceil} ((\bar{x}\backslash^{\mathcal{I}(r,k)} x' - \bar{x}\backslash^{\mathcal{I}(r,k-1)} x')_i \\
&\qquad\qquad \partial_i F((1 - z_k)(\bar{x}\backslash^{\mathcal{I}(r,k-1)} x') + z_k(\bar{x}\backslash^{\mathcal{I}(r,k)} x'))) \\
&= \frac{1}{|\widetilde{\mathcal{S}}|} \sum_{(r,z) \in \widetilde{\mathcal{S}}} \sum_{k=1}^{\lceil \frac{n}{m} \rceil} ((\bar{x}\backslash^{\mathcal{I}(r,k)} x' - \bar{x}\backslash^{\mathcal{I}(r,k-1)} x')_i \cdot 0) \\
&= 0.
\end{aligned}
\tag{15}
$$

□

## B.5 Linearity

**Definition of Linearity:** If $(\bar{x}, x', F), (\bar{x}, x', G) \in \mathcal{D}_A$ and $\alpha, \beta \in \mathbb{R}$, then $(\bar{x}, x', \alpha F + \beta G) \in \mathcal{D}_A$ and $A(\bar{x}, x', \alpha F + \beta G) = \alpha A(\bar{x}, x', F) + \beta A(\bar{x}, x', G)$.

**Lemma 8.** MBS satisfies linearity.

*Proof.* The linearity axiom of MBS can be proven from the linearity axiom of IG (Lundstrom & Razaviyayn, 2025).

$$
\begin{aligned}
A^{\mathrm{MBS}}(\bar{x}, x', \alpha F + \beta G) &= \frac{1}{n!} \sum_{r \in \mathcal{R}} \sum_{k=1}^{\lceil \frac{n}{m} \rceil} A^{\mathrm{IG}}(\bar{x}\backslash^{\mathcal{I}(r,k)} x', \bar{x}\backslash^{\mathcal{I}(r,k-1)} x', \alpha F + \beta G) \\
&= \frac{1}{n!} \sum_{r \in \mathcal{R}} \sum_{k=1}^{\lceil \frac{n}{m} \rceil} (\alpha A^{\mathrm{IG}}(\bar{x}\backslash^{\mathcal{I}(r,k)} x', \bar{x}\backslash^{\mathcal{I}(r,k-1)} x', F) \\
&\qquad\qquad + \beta A^{\mathrm{IG}}(\bar{x}\backslash^{\mathcal{I}(r,k)} x', \bar{x}\backslash^{\mathcal{I}(r,k-1)} x', G)) \\
&= \alpha A^{\mathrm{MBS}}(\bar{x}, x', F) + \beta A^{\mathrm{MBS}}(\bar{x}, x', G).
\end{aligned}
\tag{16}
$$

$\square$

**Lemma 9.** MMBS satisfies Linearity.

*Proof.* The linearity axiom of MMBS can be proven from the linearity of the partial derivative:

$$
\begin{aligned}
A_i^{\text{MMBS}}(\bar{x}, x', \alpha F + \beta G) = \frac{1}{|\widetilde{\mathcal{S}}|} \sum_{(r,z) \in \widetilde{\mathcal{S}}} \sum_{k=1}^{\lceil \frac{n}{m} \rceil} & ((\bar{x}\backslash^{\mathcal{I}(r,k)} x' - \bar{x}\backslash^{\mathcal{I}(r,k-1)} x')_i \\
& (\alpha \partial_i F((1 - z_k)(\bar{x}\backslash^{\mathcal{I}(r,k-1)} x') + z_k(\bar{x}\backslash^{\mathcal{I}(r,k)} x'))) \\
& + \beta \partial_i G((1 - z_k)(\bar{x}\backslash^{\mathcal{I}(r,k-1)} x') + z_k(\bar{x}\backslash^{\mathcal{I}(r,k)} x'))) \\
= \alpha A^{\text{MMBS}}(\bar{x}, x', F) & + \beta A^{\text{MMBS}}(\bar{x}, x', G).
\end{aligned}
\tag{17}
$$

$\square$

### B.6 Symmetry preserving

**Definition of Symmetry-Preserving:** Suppose that $(\bar{x}, x', F) \in \mathcal{D}_A$ and $i$ and $j$ are indices. Let $S_{ij}(x)$ be the function that swaps the values of $x_i$ and $x_j$. Then if $F(x) = F(S_{ij}(x))$ for any $x \in \mathcal{D}_F$, and $\bar{x} = S_{ij}(\bar{x})$ and $x' = S_{ij}(x')$, we have $A_i(\bar{x}, x', F) = A_j(\bar{x}, x', F)$.

**Lemma 10.** MBS satisfies Symmetry preserving.

*Proof.* Let $e^{(i)}$ be the $i$-th standard basis vector of $\mathbb{R}^n$. If you assume $\forall x \in \mathcal{D}_F, F(x) = F(S_{ij}(x))$, then:

$$
\partial_i F(x) = \lim_{\beta \to 0} \frac{F(x + \beta e^{(i)})) - F(x)}{\beta} = \lim_{\beta \to 0} \frac{F(S_{ij}(x) + \beta e^{(j)}) - F(S_{ij}(x))}{\beta} = \partial_j F(S_{ij}(x)).
\tag{18}
$$

By using that $\bar{x} = S_{ij}(\bar{x})$ and $x' = S_{ij}(x')$ we can derive:

$$
(\bar{x}\backslash^{\mathcal{I}(r,k)} x')_i = (\bar{x}\backslash^{\mathcal{I}(S_{ij}(r),k)} x')_j,
\tag{19}
$$

$$
\bar{x}\backslash^{\mathcal{I}(S_{ij}(r),k)} x' = S_{ij}(\bar{x}\backslash^{\mathcal{I}(r,k)} x').
\tag{20}
$$

We define $P(r, \bar{x}, x', F)$ as the sum over all IG calls for a given ordering $r \in \mathcal{R}$:

$$
P(r, \bar{x}, x', F) = \sum_{k=1}^{\lceil \frac{n}{m} \rceil} A^{\text{IG}}(\bar{x}\backslash^{\mathcal{I}(r,k)} x', \bar{x}\backslash^{\mathcal{I}(r,k-1)} x', F).
\tag{21}
$$

By using Equations 18, 19, and 20, we can prove that $P_i(r, \bar{x}, x', F) = P_j(S_{ij}(r), \bar{x}, x', F)$:

$$
\begin{aligned}
& P_i(r, \bar{x}, x', F) \\
&= \sum_{k=1}^{\lceil \frac{n}{m} \rceil} A_i^{\text{IG}}(\bar{x}\backslash^{\mathcal{I}(r,k)} x', \bar{x}\backslash^{\mathcal{I}(r,k-1)} x', F) \\
&= \sum_{k=1}^{\lceil \frac{n}{m} \rceil} \left( (\bar{x}\backslash^{\mathcal{I}(r,k)} x' - \bar{x}\backslash^{\mathcal{I}(r,k-1)} x')_i \int_{\zeta=0}^{1} \partial_i F\left( (1 - \zeta)(\bar{x}\backslash^{\mathcal{I}(r,k-1)} x') + \zeta(\bar{x}\backslash^{\mathcal{I}(r,k)} x') \right) d\zeta \right) \\
&= \sum_{k=1}^{\lceil \frac{n}{m} \rceil} \left( (\bar{x}\backslash^{\mathcal{I}(r,k)} x' - \bar{x}\backslash^{\mathcal{I}(r,k-1)} x')_i \int_{\zeta=0}^{1} \partial_j F\left( S_{ij}\left( (1 - \zeta)(\bar{x}\backslash^{\mathcal{I}(r,k-1)} x') + \zeta(\bar{x}\backslash^{\mathcal{I}(r,k)} x') \right) \right) d\zeta \right) \\
&= \sum_{k=1}^{\lceil \frac{n}{m} \rceil} \left( (\bar{x}\backslash^{\mathcal{I}(S_{ij}(r),k)} x' - \bar{x}\backslash^{\mathcal{I}(S_{ij}(r),k-1)} x')_j \int_{\zeta=0}^{1} \partial_j F\left( (1 - \zeta)(\bar{x}\backslash^{\mathcal{I}(S_{ij}(r),k-1)} x') + \zeta(\bar{x}\backslash^{\mathcal{I}(S_{ij}(r),k)} x') \right) d\zeta \right) \\
&= \sum_{k=1}^{\lceil \frac{n}{m} \rceil} A_j^{\text{IG}}(\bar{x}\backslash^{\mathcal{I}(S_{ij}(r),k)} x', \bar{x}\backslash^{\mathcal{I}(S_{ij}(r),k-1)} x', F) \\
&= P_j(S_{ij}(r), \bar{x}, x', F).
\end{aligned}
$$

$$
\tag{22}
$$

The set $\mathcal{R}$ of all possible orderings of $n$ features can be divided into two disjoint sets $\mathcal{R}_{r_i < r_j}$ and $\mathcal{R}_{r_i > r_j}$, where $\mathcal{R}_{r_i < r_j} \cup \mathcal{R}_{r_i > r_j} = \mathcal{R}$. $\mathcal{R}_{r_i < r_j}$ contains all orderings where $r_i < r_j$, and $\mathcal{R}_{r_i > r_j}$ contains all orderings where $r_i > r_j$. For every ordering $r \in \mathcal{R}_{r_i < r_j}$ there is a unique ordering $S_{ij}(r) \in \mathcal{R}_{r_i > r_j}$. Because $\mathcal{R}_{r_i < r_j} \cup \mathcal{R}_{r_i > r_j} = \mathcal{R}$, we can rewrite MBS in a way that proves that it is symmetry preserving:

$$A_i^{\mathrm{MBS}}(\bar{x}, x', F) = \frac{1}{n!} \sum_{r \in \mathcal{R}_{r_i < r_j}} P_i(r, \bar{x}, x', F) + P_i(S_{ij}(r), \bar{x}, x', F)$$

$$= \frac{1}{n!} \sum_{r \in \mathcal{R}_{r_i < r_j}} P_j(S_{ij}(r), \bar{x}, x', F) + P_j(r, \bar{x}, x', F) = A_j^{\mathrm{MBS}}(\bar{x}, x', F). \tag{23}$$

$\square$

**Lemma 11.** MMBS does not always satisfy Symmetry preserving.

*Proof.* A counteraxample is when $F(x) = (x_1 + x_2)^2$, $\bar{x}_1 = (1, 1)$, $x'_1 = (0, 0)$, and MMBS is calculated with a step size of 1 and one sample ($|\widetilde{\mathcal{S}}| = 1$), where $r = (1, 2)$ and $z = \frac{1}{2}$. In that case:

$$A_1^{\mathrm{MMBS}}(\bar{x}, x', F) = \frac{1}{|\widetilde{\mathcal{S}}|} \sum_{(r,z) \in \widetilde{\mathcal{S}}} \sum_{k=1}^{\lceil \frac{n}{m} \rceil} ((\bar{x}^{\backslash \mathcal{I}(r,k)} x' - \bar{x}^{\backslash \mathcal{I}(r,k-1)} x')_1$$

$$\partial_1 F((1 - z_k)(\bar{x}^{\backslash \mathcal{I}(r,k-1)} x') + z_k(\bar{x}^{\backslash \mathcal{I}(r,k)} x'))) \tag{24}$$

$$= (1 - 0) \cdot \partial_1 F(\frac{1}{2}(1, 0) + \frac{1}{2}(0, 0)) + (1 - 1) \cdot \partial_1 F(\frac{1}{2}(1, 1) + \frac{1}{2}(1, 0))$$

$$= 1 \cdot \partial_1 F((\frac{1}{2}, 0)) = 1,$$

$$A_2^{\mathrm{MMBS}}(\bar{x}, x', F) = \frac{1}{|\widetilde{\mathcal{S}}|} \sum_{(r,z) \in \widetilde{\mathcal{S}}} \sum_{k=1}^{\lceil \frac{n}{m} \rceil} ((\bar{x}^{\backslash \mathcal{I}(r,k)} x' - \bar{x}^{\backslash \mathcal{I}(r,k-1)} x')_2$$

$$\partial_2 F((1 - z_k)(\bar{x}^{\backslash \mathcal{I}(r,k-1)} x') + z_k(\bar{x}^{\backslash \mathcal{I}(r,k)} x'))) \tag{25}$$

$$= (0 - 0) \cdot \partial_2 F(\frac{1}{2}(1, 0) + \frac{1}{2}(0, 0)) + (1 - 0) \cdot \partial_2 F(\frac{1}{2}(1, 1) + \frac{1}{2}(1, 0))$$

$$= 1 \cdot \partial_2 F((1, \frac{1}{2})) = 3.$$

Even though $F(x) = F(S_{ij}(x))$ for any $x \in \mathcal{D}_F$, and $\bar{x} = S_{12}(\bar{x})$ and $x' = S_{12}(x')$, we have $A_i(\bar{x}, x', F) \neq A_j(\bar{x}, x', F)$. Therefore, MMBS does not satisfy Symmetry preserving. $\square$

### B.7 Non-Decreasing Positivity

**Definition of Non-Decreasing Positivity:** If $(\bar{x}, x', F) \in \mathcal{D}_A$ and $F$ is non-decreasing from $x'$ to $\bar{x}$ then $A_i(\bar{x}, x', F) \geq 0$ for every index $i$.

**Definition of Non-Decreasing (Lundstrom et al., 2022):** $F$ is non-decreasing from $x'$ to $x$ if $F(\gamma(t))$ is non-decreasing for every monotone path $\gamma(t)$ from $x'$ to $x$.

**Definition of monotone path function (Lundstrom et al., 2022; Lundstrom & Razaviyayn, 2025):** A function $\gamma(t) : [0, 1] \to \mathcal{D}_F$ is a monotone path function from $x' \in \mathbb{R}^n$ to $x \in \mathbb{R}^n$ if $\gamma(t)$ is a continuous, piecewise smooth curve from $x'$ to $x$, and $|x'_i - \gamma_i(t_1)| \leq |x'_i - \gamma_i(t_2)|$ for all indices $i$ and all $t_1, t_2 \in [0, 1]$ where $t_1 < t_2$.

**Lemma 12.** MBS satisfies non-decreasing positivity.

*Proof.* Because of the definition of non-decreasing, $F(\gamma(t))$ is non-decreasing for every monotone path $\gamma(t)$ from $x'$ to $\bar{x}$. It is also non-decreasing for every subsection of these paths. For every $k, r$, a monotone path can be constructed by connecting the points $[x', \bar{x}^{\backslash \mathcal{I}(r,k-1)} x', \bar{x}^{\backslash \mathcal{I}(r,k)} x', \bar{x}]$ using straight line segments. Therefore, $F$ is also non-decreasing from $\bar{x}^{\backslash \mathcal{I}(r,k-1)} x'$ to $\bar{x}^{\backslash \mathcal{I}(r,k)} x'$. Because of this, and because IG satisfies non-decreasing positivity (Lundstrom & Razaviyayn, 2025), $A^{\mathrm{IG}}(\bar{x}^{\backslash \mathcal{I}(r,k)} x', \bar{x}^{\backslash \mathcal{I}(r,k-1)} x', F)$ is non-negative

in every element for all $k, r$. MBS is the element-wise average of these terms, so MBS is also non-negative in every element. $\qquad\square$

**Lemma 13.** MMBS satisfies non-decreasing positivity.

*Proof.* First we define $\hat{x}_{r,k,z}$ as the point in which the gradient is calculated for any given MMBS iteration. For any given index $k$, ordering $r$, and vector $z \in \mathbb{R}^{\lceil \frac{n}{m} \rceil}$ where every element $z_l$ is in the range $[0, 1]$:

$$\hat{x}_{r,k,z} = (1 - z_k)(\bar{x}\backslash^{\mathcal{I}(r,k-1)}x') + z_k(\bar{x}\backslash^{\mathcal{I}(r,k)}x')). \tag{26}$$

For any $\hat{x}_{r,k,z}$ and index $i$, a monotone path can be constructed by connecting the points $[x', x'\backslash^{\{i\}}\hat{x}_{r,k,z}, \bar{x}\backslash^{\{i\}}\hat{x}_{r,k,z}, \bar{x}]$ using straight line segments. Therefore, when $F$ is non-decreasing, the path segment between $x'\backslash^{\{i\}}\hat{x}_{r,k,z}$ and $\bar{x}\backslash^{\{i\}}\hat{x}_{r,k,z}$ is also non-decreasing. This path segment only varies the $i$-th element, which changes from $x'_i$ to $\bar{x}_i$. All other elements are are fixed at their respective values from $\hat{x}_{r,k,z}$. $\hat{x}_{r,k,z}$ lies on this path segment, so $\partial_i F(\hat{x}_{r,k,z})$ is non-negative when $\bar{x}_i - x'_i$ is positive, and non-positive when $\bar{x}_i - x'_i$ is negative. Therefore, the product of $\bar{x}_i - x'_i$ and $\partial F_i(\hat{x}_{r,k,z})$ is non-negative. MMBS is a sum of these products, so MMBS is also non-negative. $\qquad\square$

## B.8 Affine Scale Invariance

**Definition of Affine Scale Invariance (ASI):** Suppose that $(\bar{x}, x', F) \in \mathcal{D}_A$, $\alpha, \beta \in \mathbb{R}$ with $\alpha \neq 0$, and $i$ is an index. Let $T$ be an affine transformation of element $i$, so that $T(x) := (x_1, \cdots, \alpha x_i + \beta, \cdots, x_n)$. Then we have $A(\bar{x}, x', F) = A(T(\bar{x}), T(x'), F \circ T^{-1})$.

**Lemma 14** (MBS satisfies Affine Scale Invariance.)**.** *Proof.* The *Affine Scale Invariance* of MBS can be proven by using the *Affine Scale Invariance* of IG (Lundstrom & Razaviyayn, 2025):

$$
\begin{aligned}
&A^{\mathrm{MBS}}(T(\bar{x}), T(x'), F \circ T^{-1}) \\
&= \frac{1}{n!} \sum_{r \in \mathcal{R}} \sum_{k=1}^{\lceil \frac{n}{m} \rceil} A^{\mathrm{IG}}(T(\bar{x})\backslash^{\mathcal{I}(r,k)}T(x'), T(\bar{x})\backslash^{\mathcal{I}(r,k-1)}T(x'), F \circ T^{-1}) \\
&= \frac{1}{n!} \sum_{r \in \mathcal{R}} \sum_{k=1}^{\lceil \frac{n}{m} \rceil} A^{\mathrm{IG}}(T(\bar{x}\backslash^{\mathcal{I}(r,k)}x'), T(\bar{x}\backslash^{\mathcal{I}(r,k-1)}x'), F \circ T^{-1}) \\
&= \frac{1}{n!} \sum_{r \in \mathcal{R}} \sum_{k=1}^{\lceil \frac{n}{m} \rceil} A^{\mathrm{IG}}(\bar{x}\backslash^{\mathcal{I}(r,k)}x', \bar{x}\backslash^{\mathcal{I}(r,k-1)}x', F) \\
&= A^{\mathrm{MBS}}(\bar{x}, x', F).
\end{aligned}
\tag{27}
$$

$\qquad\square$

**Lemma 15.** MMBS satisfies Affine Scale Invariance.

*Proof.* Let $c, d \in \mathbb{R}^n$ be defined as $c_l = \begin{cases} \alpha & : l = i \\ 1 & : l \neq i \end{cases}$ and $d_l = \begin{cases} \beta & : l = i \\ 0 & : l \neq i \end{cases}$, so $T_i(x) = c_i x_i + d_i$. Using this,

we show that $A_l^{\mathrm{MMBS}}(T(\bar{x}), T(x'), F \circ T^{-1}) = A_l^{\mathrm{MMBS}}(\bar{x}, x', F)$ for any index $l$:

$$
\begin{aligned}
A_l^{\mathrm{MMBS}}(T(\bar{x}), T(x'), F \circ T^{-1}) = {}& \frac{1}{|\widetilde{\mathcal{S}}|} \sum_{(r,z) \in \widetilde{\mathcal{S}}} \sum_{k=1}^{\lceil \frac{n}{m} \rceil} ((T(\bar{x}) \backslash^{\mathcal{I}(r,k)} T(x') - T(\bar{x}) \backslash^{\mathcal{I}(r,k-1)} T(x'))_l \\
& \partial_l (F \circ T^{-1})((1 - z_k)(T(\bar{x}) \backslash^{\mathcal{I}(r,k-1)} T(x')) + z_k(T(\bar{x}) \backslash^{\mathcal{I}(r,k)} T(x')))) \\
= {}& \frac{1}{|\widetilde{\mathcal{S}}|} \sum_{(r,z) \in \widetilde{\mathcal{S}}} \sum_{k=1}^{\lceil \frac{n}{m} \rceil} ((T(\bar{x} \backslash^{\mathcal{I}(r,k)} x') - T(\bar{x} \backslash^{\mathcal{I}(r,k-1)} x'))_l \\
& \partial_l (F \circ T^{-1})((1 - z_k) T(\bar{x} \backslash^{\mathcal{I}(r,k-1)} x') + z_k T(\bar{x} \backslash^{\mathcal{I}(r,k)} x'))) \\
= {}& \frac{1}{|\widetilde{\mathcal{S}}|} \sum_{(r,z) \in \widetilde{\mathcal{S}}} \sum_{k=1}^{\lceil \frac{n}{m} \rceil} (((c_l(\bar{x} \backslash^{\mathcal{I}(r,k)} x')_l + d_l) - (c_l(\bar{x} \backslash^{\mathcal{I}(r,k-1)} x')_l + d_l)) \\
& \partial_l (F \circ T^{-1})(T((1 - z_k) \bar{x} \backslash^{\mathcal{I}(r,k-1)} x' + z_k \bar{x} \backslash^{\mathcal{I}(r,k)} x'))) \\
= {}& \frac{1}{|\widetilde{\mathcal{S}}|} \sum_{(r,z) \in \widetilde{\mathcal{S}}} \sum_{k=1}^{\lceil \frac{n}{m} \rceil} (c_l(\bar{x} \backslash^{\mathcal{I}(r,k)} x' - \bar{x} \backslash^{\mathcal{I}(r,k-1)} x')_l \\
& \frac{1}{c_l} \partial_l F(T^{-1}(T((1 - z_k) \bar{x} \backslash^{\mathcal{I}(r,k-1)} x' + z_k \bar{x} \backslash^{\mathcal{I}(r,k)} x')))) \\
= {}& \frac{1}{|\widetilde{\mathcal{S}}|} \sum_{(r,z) \in \widetilde{\mathcal{S}}} \sum_{k=1}^{\lceil \frac{n}{m} \rceil} ((\bar{x} \backslash^{\mathcal{I}(r,k)} x' - \bar{x} \backslash^{\mathcal{I}(r,k-1)} x')_l \\
& \partial_l F((1 - z_k) \bar{x} \backslash^{\mathcal{I}(r,k-1)} x' + z_k \bar{x} \backslash^{\mathcal{I}(r,k)} x')).
\end{aligned}
\tag{28}
$$

$\square$

# C   Convergence plots for the ImageNet networks

Calculating MBShap at a high number of iterations, so that it converges almost completely, is infeasible for the ImageNet networks. Therefore, the convergence behavior of MMBS on the ImageNet networks was investigated using two different reference images that provide complementary information. The first baseline was 10,000 iterations of MMBS with 128 steps, which was used to check the convergence behavior with respect to the number of iterations and steps. The second reference image was MBShap at a low number of iterations (100 for ResNet and 30 for the Vision Transformer), which was used to check whether the results approximated BShap. Despite using only a low number of iterations, the computation time of MBShap was still very high, so these experiments were performed on 16 images on the ResNet network and 8 images on the Visition Transformer network. The results are shown in Figures C.1 and C.2.

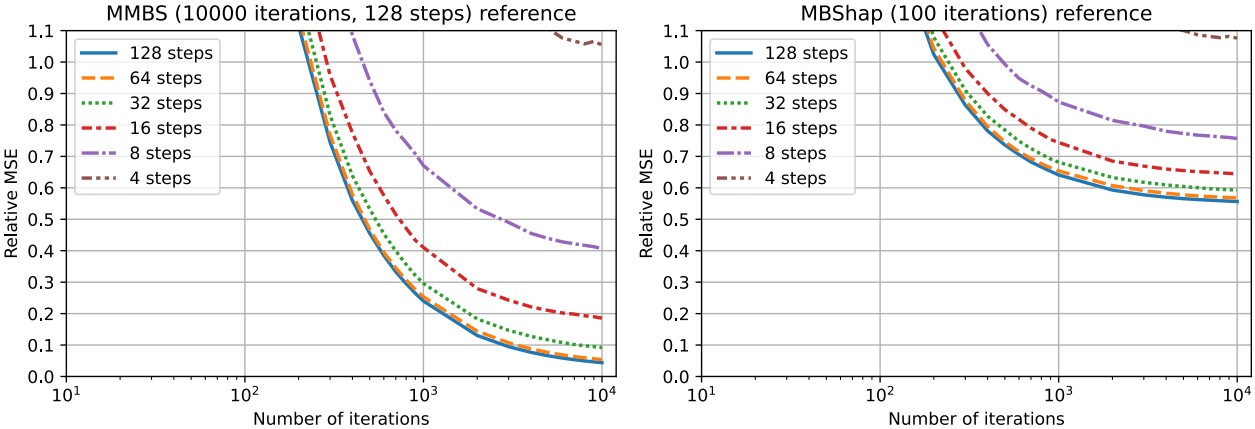

Figure C.1: Convergence of MMBS on the ResNet50 transformer network as measured by the relative MSE calculated for two reference images: 10.000 iterations of MMBS with 128 steps, and MBShap with 100 iterations. MMBS with 1 or 2 steps had a relative MSE that was consistently higher than 1.1, so it was not included in the plot.

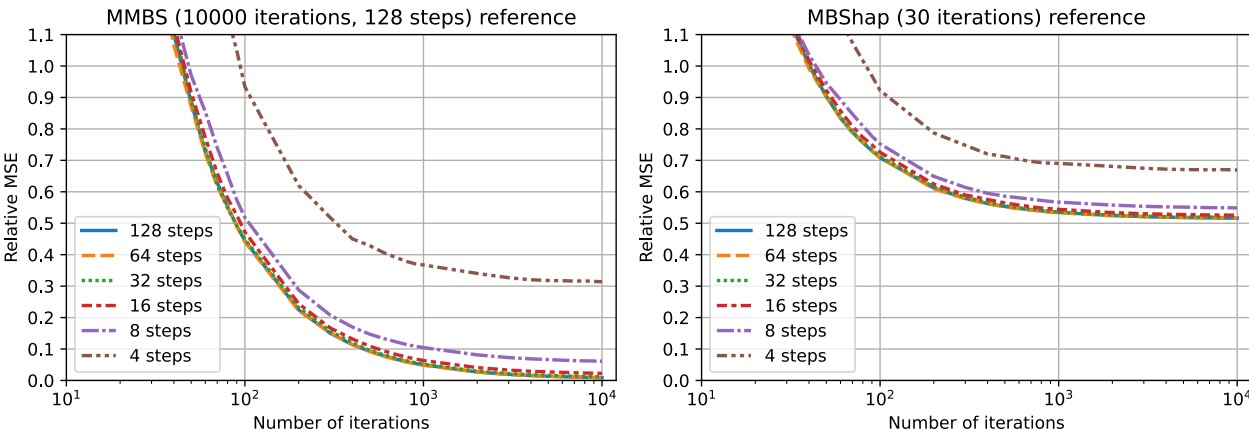

Figure C.2: Convergence of MMBS on the vision transformer network as measured by the relative MSE calculated for two reference images: 10.000 iterations of MMBS with 128 steps, and MBShap with 30 iterations. MMBS with 1 or 2 steps had a relative MSE that was consistently higher than 1.1, so it was not included in the plot.

For both networks, the relative MSE between the two MMBS attribution maps with 128 steps at 10,000 iterations was close to zero, suggesting that they were almost fully converged at 10,000 iterations. Moreover, the relative MSE was very similar when the step size was 16 or higher on the Vision Transformer, or 64 or higher on the ResNet. This is promising because it suggests that further increasing the number of steps is

not necessary to approximate BShap. The relative MSE curves that used MBShap with a low number of iterations show a systematic error, but this was expected because the reference was not fully converged.

Additionally, MMBS with 8 steps and MBShap were calculated for up to 160 iterations, on a single image, for both ImageNet networks, and this is plotted in Figure C.3. It shows that even when MMBS and MBShap are not fully converged, they already produce visually similar images.

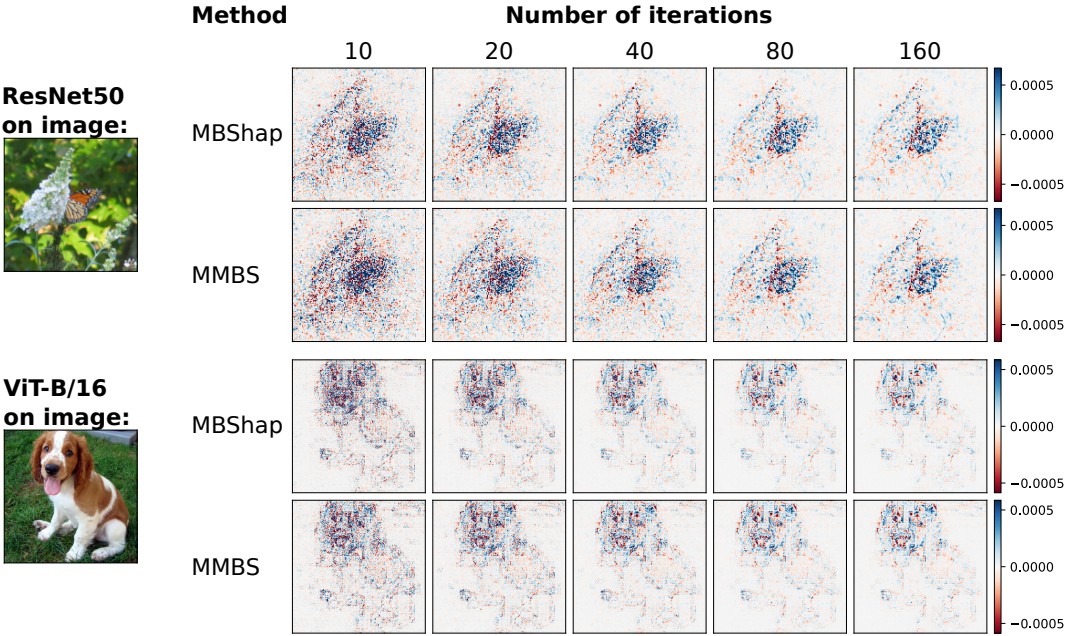

Figure C.3: Example of the convergence of MBShap and MMBS attribution maps on a ResNet and a Vision Transformer network. MMBS produces visually similar results as MBShap at the same number of iterations.

# D Sensitivity of the AUDC to the number of deletion curve steps

The AUDC scores in the main text of this paper were approximated from 200 equally spaced points along the deletion curve. To measure the sensitivity of the AUDC to the number of points, the AUDC was also calculated for other numbers of points (50, 100, 200, 400, 800, 1600). On the ImageNet network, the same methods with the same settings were tested as in the experiment in Section 4.4, but to limit the computation time of this experiment, fewer images were used. Attribution maps were calculated for the first image from the validation set of every 10th class, resulting in 100 attribution maps. On the Vision Transformer network, attribution maps were calculated for the same 100 images, but to limit the computation time further, only the two best-performing methods from Table 2 were used (MMBS and GIG (paper) + SmoothGrad). In all cases, increasing the number of AUDC steps beyond 200 had only a very small effect on the calculated AUDC score.

Table D.1: Mean AUDC at different numbers of points used in calculating the deletion curve. These results were calculated using 100 images on the ImageNet ResNet network.

| Method | Mean AUDC with ... deletion curve points | | | | | | Difference between 200 and 1600 points |
|---|---|---|---|---|---|---|---|
| | 50 | 100 | 200 | 400 | 800 | 1600 | |
| MMBS | 0.017 | 0.016 | 0.015 | 0.015 | 0.015 | 0.015 | 0.00021 |
| MMBS + SG | 0.024 | 0.023 | 0.023 | 0.023 | 0.023 | 0.023 | 0.00006 |
| IG | 0.063 | 0.063 | 0.063 | 0.063 | 0.063 | 0.063 | 0.00002 |
| IG + SG | 0.038 | 0.037 | 0.037 | 0.037 | 0.037 | 0.037 | 0.00006 |
| GIG (paper) | 0.107 | 0.106 | 0.105 | 0.105 | 0.105 | 0.105 | 0.00040 |
| GIG (paper) + SG | 0.038 | 0.037 | 0.037 | 0.037 | 0.037 | 0.037 | 0.00002 |
| GIG (Saliency) | 0.049 | 0.048 | 0.048 | 0.048 | 0.048 | 0.048 | 0.00019 |
| GIG (Saliency) + SG | 0.031 | 0.030 | 0.030 | 0.030 | 0.030 | 0.030 | 0.00005 |
| XRAI (B + W) | 0.173 | 0.173 | 0.173 | 0.173 | 0.173 | 0.173 | 0.00010 |
| XRAI (zero) | 0.179 | 0.179 | 0.178 | 0.178 | 0.178 | 0.178 | 0.00014 |

Table D.2: Mean AUDC at different numbers of points used in calculating the deletion curve. These results were calculated using 100 images on the ImageNet Vision Transformer network.

| Method | Mean AUDC with ... deletion curve points | | | | | | Difference between 200 and 1600 points |
|---|---|---|---|---|---|---|---|
| | 50 | 100 | 200 | 400 | 800 | 1600 | |
| MMBS | 0.023 | 0.021 | 0.021 | 0.020 | 0.020 | 0.020 | 0.00038 |
| GIG (paper) + SG | 0.023 | 0.021 | 0.021 | 0.021 | 0.021 | 0.021 | 0.00021 |

