# OpenReview forum: "Monte Carlo Multi-Feature Baseline Shapley (MMBS): An axiomatic attribution method for fine-grained explanations of image classification networks"
_TMLR — Accepted by TMLR_

### Review · Reviewer_haht · 2026-05-07

**Summary Of Contributions:**

# Summary

This paper proposes a new method, called *Multi-Feature Baseline Shapley (MBS)* for providing explanations of neural network predictions over images. Their proposed methods generalize previous work, especially *Integrated Gradients (IG)* and *Baseline Shapley (BShap)* under a common framework, depending on the step size parameter of their method. Through a series of experiments, the authors claim that their method (i) runs faster than BShap and (ii) is more faithful, according to the AUDC metric, to the underlying neural network's actual decision process.

# Strong Points

- The reported speedups (Figs 2 and 3) are massive
- The acquired speedups seem to be obtained while having better AUDC
- Experiments are comprehensive

# Weak Points

- (Minor, stylitic) The way the axioms are layed  out in Section 3.3 is a bit unorthodox. I would suggest the authors to use a centralized Definition environment (Definition 1 (MBS Axioms). ...) then provide at least a formal statement with the hypothesis they need to assume for the results in Appendix B to hold, in the main paper (meaning, a theorem or proposition statement).

Now, concerning the main claims made in the paper, I have a couple of other weaknesses

- In Figure 2, the authors provide an unclear description about the algorithms convergence, for instance, they mention,

> MBShap converged at a fairly low number of samples (around 256). MMBS closely approximated MBSHap at a fairly low number of steps (around 8).

It is not clear which criterion is used for verifying convergence,  and they don't provide a clear enough criterion of covergence. The same problem is also present in Fig. 3

- In Figure 3, the avg. time per iteration is only reported for a couple of images. It would be nice to have a more robust statistical analysis over a set of $n$ images, reporting mean $\pm$ std.

**Additional Comments:**

N/A

**Audience:**

Yes

**Audience Explanation:**

Explainability is an active field of ML hence of interest to TMLR.

**Claims And Evidence:**

Yes

**Claims Explanation:**

To be clear, I'd say *"Partially Yes"*. There are claims that need refinement (see weakness in the summary and the requested changes below).

**Requested Changes:**

Here is a list of changes,

1) I think it would be much clearer if the authors centralize the 8 axioms under a single definition, and provide a numbering for the axioms (for short identification). A formal statement that MMBS satisfies the 8 axioms is also necessary.

2) The authors should make an effort to make some unclear statements more quantifiable and statistical relevant, especially,

2.1) Specify the criterion for convergence of the algorithms,

2.2) Track runtime over a set of images (say, 16), and report the average and standard deviation over the runs

---

> ### Author Response · Authors · 2026-05-28
> **Reply to review**
>
> Thank you for your insightful review. We believe it has helped improve the paper substantially.
>
> From your review, we identified three main areas of improvement: 1.) Improve the structure and formality of the mathematical definitions and propositions. 2.) Specify the criterion for convergence of the algorithms, and 3.) Track runtime over a set of images (say, 16), and report the average and standard deviation over the runs. We reply to each of these comments below.
>
> *Comment 1:* "Improve the structure and formality of the mathematical definitions and propositions."
> *Reply 1:* As suggested, we have numbered the axioms and put the main mathematical findings in proposition statements. Moreover, the proofs in Appendix B are now written as numbered Lemmas, making them easier to reference. We also tried putting all axioms directly under each other in a single definition, but we think it is more readable to have the explanations about the axioms directly after their definitions, so we maintained that from the previous version of the paper.
>
> *Comment 2:* "Specify the criterion for convergence of the algorithms."
> *Reply 2:* We completely rewrote Section 4.2 and added Appendix C to report the convergence in a more precisely specified way, and over more images.
>
> *Comment 3:* "Track runtime over a set of images (say, 16), and report the average and standard deviation over the runs."
> *Reply 3:* We have added Section 4.3 to report on the runtime of MMBS. We measured the runtime of MMBS at different step sizes and MBShap for 50 images of each dataset, and we report the average and standard deviation in Table 1.

---

> > ### Comment · Reviewer_haht · 2026-06-03
> > **Reply to authors**
> >
> > Dear authors,
> >
> > Thank you for making the changes I requested.
> >
> > The current state of the manuscript is much better.
> >
> > I have one minor suggestion for the new convergence figures (which are super interesting, BTW). I think the authors could do log-log plots (log on the x and y scales) so that we can see the trends and better differentiate between the curves. I'd say this change is optional, if the authors feel the current figures are clear enough, or that the log-log plots don't offer much more information, I am happy with the current state of the manuscript..

---

> ### Author Response · Authors · 2026-06-19
> **Update to the plots**
>
> We agree that the plots can be improved. We tried making log-log plots, but found them less visually intuitive, and found it harder to read the value of a curve precisely. Therefore, we've continued using semilog plots, but made some changes to make the trends clearer, and the curves easier to differentiate:
> - We have narrowed the range of displayed relative MSE values to [0, 1.1], so there is more space between the curves, and because values above 1 are considered bad anyway.
> - We have made the plots slightly taller, so there is more space between the curves.
> - We have added more gridlines so that every multiple of 0.1 has a gridline, making the values of curves easier to read.
>
> We have uploaded a revision of the paper with the updated plots.

---

> > ### Comment · Reviewer_haht · 2026-06-19
> > **Response to authors**
> >
> > Dear authors,
> >
> > Thank you for trying out the log-log plots. I am happy with the final state of the manuscript.

---

### Review · Reviewer_vtmB · 2026-05-11

**Summary Of Contributions:**

This work proposed Multi-Feature Baseline Shapley (MBS) method and its Monte Carlo variant, MMBS, to calculate the attribution map for explainable machine learning. The proposed method is a generalization of IG and BShap by introducing a stepsize parameter. They further proved that the proposed methods satisfy certain axioms. In the experiments, they show that the proposed methods achieved significant speedup over existing methods, and better performances in terms of AUDC compared to existing methods.

**Audience:**

Yes

**Audience Explanation:**

The main topic of the work is related to explainable machine learnining, which is an important topic in ML. The main idea of the proposed method is promising, and the empirical performance is favorable compared to several existing methods. So the findings should be interesting to some TMLR's audience.

**Broader Impact Concerns:**

No concerns.

**Claims And Evidence:**

Yes

**Claims Explanation:**

This work presented a new method for attribution map calculation, the motivation and design of the method are reasonable, the empirical results are promising.

**Requested Changes:**

Some potential issues:
1. Regarding the speedup claims in Section 4.2, you compare their performances in a vague metric, you mentioned "both methods converge to very similar images", which is more like a visual comparison, and not convincing enough. I suggest a more quantitative comparison in terms of the computation time, e.g., report quality-versus-runtime or time-to-target-accuracy curves if possible.
2. MMBS is an unbiased estimator of MBS, but not that of BShap (for general $m$) as you mentioned, so there could be a systematic approximation error, so the statement "MMBS as a fast approximation of BShap" should be further double check, is there any further quantitative comparison results on the two methods in terms of their performances?
3. Does MMBS satisfy those axioms (exactly or in expectation or in some other ways)? Currently it is not discussed in the work, if not satisfied, how should we understand the feasibility of MMBS?

---

> ### Comment · Action_Editor_fj8L · 2026-05-12
> **Claims and evidence**
>
> Dear reviewer vtmB,
>
> in your review when discussing the question 'Are the claims made in the submission supported by accurate, convincing and clear evidence?' you explain your positive answer through the novelty of the method and reasonable motivation and design. These are certainly important qualities of the paper, nevertheless this questions is directed and the claims and provided evidence. Please see here for detailed explanation of the TMLR criteria (which are admittedly somewhat special): https://jmlr.org/tmlr/acceptance-criteria.html
>
> Could you please update your justification in this part of the paper in view of the "claim-evidence" viewpoint? In particular, are all claims sufficiently supported and the overall paper is technically sound and clear? Or are there any gaps the authors shall address.
>
> Thanks a lot,
> your AE

---

> ### Author Response · Authors · 2026-05-28
> **Reply to the review**
>
> Thank you for your insightful review. We believe it has helped improve the paper substantially.
>
> Below, we provide a point-by-point response to your requested changes.
>
> *Comment 1:* "Regarding the speedup claims in Section 4.2, you compare their performances in a vague metric, you mentioned "both methods converge to very similar images", which is more like a visual comparison, and not convincing enough. I suggest a more quantitative comparison in terms of the computation time, e.g., report quality-versus-runtime or time-to-target-accuracy curves if possible."
> *Reply 1:* We have added Table 1, which provides more extensive measurements of the runtime of MMBS and MBShap. We also added Figures 2, C1, and C2 that relate the number of iterations of MMBS at different numbers of steps to (approximate) measures of convergence. Together, they provide an indication of how long it takes to reach a certain level of convergence. We have removed claims from the paper that state a specific speedup number.
>
> *Comment 2:* "MMBS is an unbiased estimator of MBS, but not that of BShap (for general ) as you mentioned, so there could be a systematic approximation error, so the statement "MMBS as a fast approximation of BShap" should be further double check, is there any further quantitative comparison results on the two methods in terms of their performances?"
> *Reply 2:* We completely rewrote Section 4.2 and added Appendix C to report the convergence in a more precisely specified way, and over more images.
>
> *Comment 3:* "Does MMBS satisfy those axioms (exactly or in expectation or in some other ways)? Currently it is not discussed in the work, if not satisfied, how should we understand the feasibility of MMBS?"
> *Reply 3:* We have added Propositions 4 and 5, and Lemmas 3, 5, 7, 9, 11, 13, and 15 to be more precise in specifying the axiomatic properties of MMBS. In expectation, MMBS satisfies the same properties as MBS, because MMBS is an unbiased estimator of MBS. In general, MMBS satisfies 5 out of the 8 axioms. Given that MMBS appears to approximate BShap closely at a fairly low number of steps and iterations, it should be possible to obtain results in practice that are close to satisfying all axioms.
> It should be noted that IG shows similar behavior. The proofs that show that IG satisfies axioms assume that the integral can be calculated perfectly, but in practice, the integral is approximated numerically. I’m not aware of any prior work that proves the axiomatic properties of IG when the integral is numerically approximated. However, the counterexamples in Lemmas 3 and 5 also work as counterexamples for IG when the integral is approximated with just one sample.

---

### Review · Reviewer_ouBb · 2026-05-16

**Summary Of Contributions:**

In this work, the authors introduce a new method, Multi-Feature Baseline Shapley (MBS), for feature attribution in deep neural networks used for image classification. MBS is a generalization of the widely used Integrated Gradients (IG) attribution method. MBS computes IG over a fixed number of “steps” or subsets of features for each possible ordering, with the size of the step being a hyperparameter. Since the number of possible feature orderings grows factorially with the number of features, the authors simultaneously introduce a Monte Carlo approximation of MBS, dubbed Monte Carlo MBS (MMBS), that computes attribution maps over a randomly sampled set of orderings. They show that MMBS is asymptotically unbiased and empirically performs well, converging to the same attribution map as MBS for a relatively modest number (~256) of sampled orderings. The authors further evaluate their method on three benchmark datasets, Fashion MNIST, ImageNet Resnet, and ImageNet ViT, using the Area Under the Deletion Curve (AUDC) as a quantitative metric for comparison against existing methods such as IG, among others.

### Strengths

- The paper is generally well written and the results are clearly presented

- The methodological contribution appears to be novel and is well explained

- Numerical experiments are relatively thorough and clearly demonstrate both the faithfulness of the MMBS approximation and its competitiveness with existing attribution methods under a specific choice of metric


### Weaknesses

- The motivation for MBS and why we should want to apply BShap to images is not clear; furthermore, the authors never clearly define the specific attribution goal their method aims to achieve

- Both the qualitative and quantitative evidence in favor of the proposed method are suggestive but somewhat weak due to a fairly considerable amount of ambiguity and uncertainty

- There is a lack of engagement with recent literature on fundamental challenges in NN attribution

- The experimental analysis is limited to three very common benchmark datasets and lacks clear goals or standards by which the explanations produced by the attribution maps could be evaluated

**Additional Comments:**

I checked the proofs for general sensibility and did not spot any obvious errors. Most of the proofs are fairly straightforward extensions of the results for the existing IG method. I would nevertheless encourage the editor to consult a referee who is more experienced within the field of NN attribution theory to do a more thorough assessment.

### References

[1] R. Geirhos, R. S. Zimmermann, B. Bilodeau, W. Brendel, and B. Kim, “Don’t trust your eyes: on the (un)reliability of feature visualizations,” in *Proceedings of the 41st International Conference on Machine Learning*, in ICML’24, vol. 235. Vienna, Austria: JMLR.org, Jul. 2024, pp. 15294–15330.

[2] B. Bilodeau, N. Jaques, P. W. Koh, and B. Kim, “Impossibility theorems for feature attribution,” *Proceedings of the National Academy of Sciences*, vol. 121, no. 2, p. e2304406120, Jan. 2024, doi: [10.1073/pnas.2304406120](https://doi.org/10.1073/pnas.2304406120).

[3] A. Ghorbani, A. Abid, and J. Zou, “Interpretation of Neural Networks Is Fragile,” *Proceedings of the AAAI Conference on Artificial Intelligence*, vol. 33, no. 01, pp. 3681–3688, Jul. 2019, doi: [10.1609/aaai.v33i01.33013681](https://doi.org/10.1609/aaai.v33i01.33013681).

[4] I. E. Kumar, S. Venkatasubramanian, C. Scheidegger, and S. Friedler, “Problems with Shapley-value-based explanations as feature importance measures,” in *Proceedings of the 37th International Conference on Machine Learning*, PMLR, Nov. 2020, pp. 5491–5500. Accessed: May 12, 2026. [Online]. Available: [https://proceedings.mlr.press/v119/kumar20e.html](https://proceedings.mlr.press/v119/kumar20e.html)

[5] D. Janzing, L. Minorics, and P. Bloebaum, “Feature relevance quantification in explainable AI: A causal problem,” in *Proceedings of the Twenty Third International Conference on Artificial Intelligence and Statistics*, PMLR, Jun. 2020, pp. 2907–2916. Accessed: May 13, 2026. [Online]. Available: [https://proceedings.mlr.press/v108/janzing20a.html](https://proceedings.mlr.press/v108/janzing20a.html)

[6] J. Marques-Silva, “Disproving XAI Myths with Formal Methods – Initial Results,” in *2023 27th International Conference on Engineering of Complex Computer Systems (ICECCS)*, Jun. 2023, pp. 12–21. doi: [10.1109/ICECCS59891.2023.00012](https://doi.org/10.1109/ICECCS59891.2023.00012).

[8] M. Sundararajan and A. Najmi, “The Many Shapley Values for Model Explanation,” in *Proceedings of the 37th International Conference on Machine Learning*, PMLR, Nov. 2020, pp. 9269–9278. Accessed: May 16, 2026. [Online]. Available: [https://proceedings.mlr.press/v119/sundararajan20b.html](https://proceedings.mlr.press/v119/sundararajan20b.html)

**Audience:**

Yes

**Audience Explanation:**

The proposed method constitutes a novel (as far as I am aware) methodological development for feature attribution in some types of neural networks and thus is well within the scope of TMLR.

**Broader Impact Concerns:**

Like virtually all “XAI” methods, the method proposed here by the authors could end up being used in ethically sensitive application domains, e.g. to provide explanations for neural networks that are used to make operational decisions in medicine, law, content moderation, or even armed conflicts. Thus, some care should be taken to underscore the limitations of both the authors’ method and other attribution methods more generally. As such, I am somewhat concerned by the authors’ lack of engagement with the literature on the topic of fundamental challenges in XAI and NN attribution. There have been numerous papers in recent years criticizing both NN attribution methods (like IG) [1-3] and XAI methods generally [4-6]. Of particular relevance to the present work is [2], which claims to prove that attribution methods like IG (and presumably also MBS) adhering to the completeness and linearity axioms are no better than random guessing. The authors should clarify whether or not this result applies to their method and what considerations should be taken into account by a practitioner applying their method to a “real world” model.

The authors might object that these more fundamental issues regarding the practical application of their (and others’) methods are outside of the scope of this paper, which is framed primarily as a technical contribution. I will leave this decision to the AE. However, I would point out that methods are usually presented with the intention of being used, otherwise they serve no real purpose. As such, I think the authors should make at least some effort to address these challenges.

**Claims And Evidence:**

No

**Claims Explanation:**

From a purely technical perspective, the authors’ central claims are only weakly supported by the presented evidence. The proposed method does appear to be competitive with other methods under the chosen AUDC metric, and the evidence that MMBS provides a faithful and efficient approximation of MBS is also reasonably convincing. However, there are also some aspects of the analysis which should be questioned.

One obvious issue is that the uncertainties (presumably standard deviations, but this information is missing) in Tables 1 and 2 are quite large, in many cases on the same order of magnitude as the estimate of the mean AUDC. Taking Table 1 as an example, the top three contenders for Fashion MNIST are MMBS (0.051 ± 0.052), MMBS + SG (0.054 ± 0.049), and GIG (Saliency) + SG (0.078 ± 0.067). Assuming that the stated uncertainties are standard deviations or errors, these results are statistically almost indistinguishable. The same can be said for many of the other methods and datasets. Furthermore, the AUDC is a nonnegative metric, so reporting symmetric error bars may be misleading. The authors should consider using asymmetric error bars based on quantiles, if the AUDC score distributions are strongly right-skewed.

Another issue is in the qualitative interpretation of the attribution maps, e.g. in Figure 1. The authors claim that MMBS performs better than the other methods because it correctly identifies the dalmatians’ spots. This claim is dubious at best. While it is true that the attribution maps show positive attribution on some of the black spots, they also show negative attribution for others. Furthermore, the attribution map in the second row produced by MMBS appears to assign very strong attribution weight to the dog’s eye. This isn’t necessarily wrong, i.e. it could be that dalmatians have unique eyes that serve as good predictors which is learned by the model, but it also could equally well be that either the attribution method or underlying model “mistakes” the eye for a black spot. The comparison with XRAI is also questionable since this method is designed to produce attribution maps highlighting larger regions of the image, thus expecting it to highlight fine-grained details, like the dogs’ spots, is not reasonable.

Finally, I would question the suitability of the AUDC as the sole metric for quantitative comparison of the attribution methods. If I understand correctly, the AUDC would be minimized in the case where the network output drops to zero after removing a single feature (i.e. k=1), fully attributing the prediction to that single feature. While that may tell us something about what the network learned, I do not see how this tells us anything about how accurate the attribution method is with respect to the “true” behavior of the model.

**Requested Changes:**

### General comments

- The authors should clarify how the error bars in Tables 1 and 2 are computed.

- Since the AUDC is bounded from below, the authors should consider using asymmetric error bars, or add a comment clarifying how the symmetric standard errors should be interpreted.

- The literature review should be expanded to address general concerns with and challenges in the robustness of attribution methods such as  those proposed in this work.

- The introduction should make more clear (a) the motivation for using BShap (this is also not clear in the original paper [8]) and thus why MBS/MMBS are necessary, and (b) the specific attribution problems that the authors are trying to solve. Currently the authors seem to regard attribution as a somewhat abstract mathematical problem of computing the sensitivity of the response to input features; however, a key finding of [2] is that end-tasks must be well defined for attribution methods to be at all credible.


### Detailed comments

1\. “Dalmatian dogs can be distinguished from other dog breeds by their black spots, so you would expect attribution maps to highlight the regions that contain spots. However, multiple attribution map methods in Figure 1 produced results that are too smooth or too noisy to highlight the spots.” (p.1)

This reasoning is faulty. The attribution method aims to provide an accurate explanation of the model prediction; whether or not this prediction is made based on the same criterion used by a human is an orthogonal question. In order to evaluate the attribution method on this basis, we would need to first have established that the model in fact does make the classification based on the dalmatian’s spots. This results in circular reasoning since the whole point of the attribution method is to verify such behavior in the first place. This statement and the corresponding qualitative comparison in Figure 1 should therefore be removed or revised to correctly distinguish between these questions.

2\. “Baseline Shapley (BShap) is another axiomatic attribution method that satisfies many of the same axioms as IG (Sundararajan & Najmi, 2020).” (p. 2)

Here I would expect some explanation of why BShap represents an improvement over other methods and why we should want to apply it to images in the first place.

3\. “MMBS also produces results that are significantly less visually noisy than IG (Figure 1)” (p. 2)

This is an anecdotal and subjective claim. Clarify what is meant here by “less visually noisy” and describe how and why this conclusion was drawn.

4\. “IG attribution maps are calculated by integrating the gradients over the straight-line path between a baseline and the neural network input.” (p. 3)

The term “baseline” was not defined prior to this point and could be easily confused with the more general use of the term to refer to a baseline method. This should be clarified.

5\. “BShap satisfies many of the same axioms as IG, but there are also some differences (Sundararajan & Najmi, 2020; Friedman & Moulin, 1999; Sprumont, 1998).”

What differences? Be specific and list them here.

6\. Section 3.1 - Notation

The set of NN functions $\mathcal{F}^2$ is never formally defined. It would appear from context that only NNs with scalar outputs are considered. This should be clarified.

7\. “MBS is equal to IG when m = n.” (p. 4)

Does IG also sum over all possible rankings $r \in \mathcal{R}$? Setting $m = n$ does not eliminate this sum from Eq. 1.

8\. “Linearity and Symmetry-Preserving ensure that the attribution map preserves certain properties of the network.” (p. 5)

What properties? Be specific.

9\. “To lower the computation time, we approximated Equation 8 by sampling 200 equally spaced points along the deletion curve and assuming linear changes between these points.”

Please provide a simple sensitivity check that demonstrates the error incurred by this approximation of the metric.

10\. Please clarify the usage of the terms “iterations” and “steps”. As far as I can tell, iterations here refers to the number of rankings sampled and steps refers to the number of feature batches considered in MBS.

11\. “From the Fashion MNIST dataset, 1000 randomly sampled images from the test set were used, and from the ImageNet dataset, the first image from each class in the validation set was used, also resulting in 1000 images.” (p. 8)

Why does the procedure differ between these two datasets?

12\. “when the network outcome on the input image was at least 0.2 and on the baseline image was at most 0.05”

Please explain how these thresholds were determined.

13\. “Using this reasoning, we can derive that MBS and BShap do not satisfy the axioms of Proportionality and Symmetric Monotonicity.”

What are the implications of this? Why should the reader care?

---

> ### Author Response · Authors · 2026-05-28
> **Reply to the review (part 1)**
>
> Thank you for your insightful review. We believe it has helped improve the paper substantially.
>
> Below, we provide a point-by-point response to your requested changes.
>
> ## Replies to the general comments
> *Comments 1 and 2:* The authors should clarify how the error bars in Tables 1 and 2 are computed. Since the AUDC is bounded from below, the authors should consider using asymmetric error bars, or add a comment clarifying how the symmetric standard errors should be interpreted.
> *Reply 1 and 2:* We agree that we should have specified this. Earlier, the “error bars” in Tables 1 and 2 were the standard deviation. We have changed this to be the 5th and 95th percentile.
>
> *Comment 3:* The literature review should be expanded to address general concerns with and challenges in the robustness of attribution methods such as those proposed in this work.
> *Reply 3:* We have added an additional paragraph in the related work section that reviews studies that critique XAI attribution maps. It cites multiple of the papers that were provided in your Broader Impact Concerns section.
>
> *Comment 4:* The introduction should make more clear (a) the motivation for using BShap (this is also not clear in the original paper [8]) and thus why MBS/MMBS are necessary, and (b) the specific attribution problems that the authors are trying to solve. Currently the authors seem to regard attribution as a somewhat abstract mathematical problem of computing the sensitivity of the response to input features; however, a key finding of [2] is that end-tasks must be well defined for attribution methods to be at all credible.
> *Reply 4a:* Before writing this paper, it was not clear whether BShap would be better than IG. In the paper that introduces BShap, the authors do not express a clear preference for either of them, and even provide an argument why IG might be preferable (Sundararajan & Najmi, 2020)(remark 4.6). However, the fact that BShap satisfies many of the same axioms makes the BShap method interesting to investigate, as the axiomatic properties are the main argument for using IG. We modified the introduction to clarify this.
> *Reply 4b:* We do not define a specific task that this method could be used for, because that would deviate too much from the original version of the paper. However, we agree with the key finding of [2] and have added in the related work that potential users of our method should be specific in what they aim to achieve. We also added that they should check whether the MMBS’s axiomatic properties are suitable for that goal.

---

> > ### Author Response · Authors · 2026-05-28
> > **Reply to the review (part 2)**
> >
> > ## Replies to the detailed comments
> > *Reply to 1:* We agree that the reasoning was faulty, and we’ve removed those statements from the introduction.
> > *Reply to 2:* See the reply to general comment 4a
> > *Reply to 3:* We have moved our description of visually noisy from the related work to the introduction. We believe it’s a fairly intuitive concept, and reducing visual noisiness has been one of the motivations behind existing attribution methods (Kapishnikov et al., 2021, Smilkov et al., 2017). However, we agree that our paper does not contain experiments where the reduction in visual noisiness is measured explicitly, so we have removed the claim that MMBS produces less visually noisy results from the last paragraph of the introduction.
> > *Reply to 4:* We have added an explanation of our usage of the term baseline.
> > *Reply to 5:* We have edited this section to be more specific. Moreover, Section 5.1 of the discussion further discusses this topic.
> > *Reply to 6:* We have clarified that attribution maps are only calculated for one output of the network. The set of NN functions is defined in words in Section 3.1. For a more formal definition, we refer to (Lundstrom & Razaviyayn 2025), which we have clarified in the text.
> > *Reply to 7:* Yes, MBS with m = n averages over all rankings, but for every ranking, the term is the same and equal to IG. Therefore, it yields the same result as IG. We’ve explained this more clearly in the revised version of the paper.
> > *Reply to 8:* We have explained this more clearly in the revised version of the paper.
> > *Reply to 9:* In Appendix D, we include a sensitivity check that shows that increasing the number of steps beyond 200 has little effect on the calculated AUDC scores.
> > *Reply to 10:* I have added a sentence explaining this in Section 4.2.
> > *Reply to 11:* These procedures differ mainly because of practical reasons. We reused the data-loading functionality from the code of the MNSIT network, which already shuffled the data. For ImageNet, we wrote the loader ourselves. Given the large number of classes of ImageNet, we aimed to maximize the variation in classes by using one image from every class. Given the low number of classes in Fashion MNIST (10) compared to the number of used images (1000), we didn’t deem it necessary to ensure that each class was sampled an equal number of times. We have added to the text that one image from every class was used “to ensure that images from all classes were included”.
> > *Reply to 12:* These values were chosen by the authors to avoid cases where the deletion curve is increasing or where the evaluation baseline is not neutral. To avoid manually set thresholds and to exclude fewer images, we have changed this criterion to only exclude images where $F(\bar{x}) < F(x’’)$. This excludes only a few images for each network.
> > *Reply to 13:* Whether this is relevant to the reader depends on the attribution task of the reader. In the added paragraph of the related work, we emphasize that users should be clear about what they want to visualize with their attribution map, and that axiomatic guarantees can help in specifying how a method behaves.
> >
> > ## Additional reply
> > *Comment:* "One obvious issue is that the uncertainties (presumably standard deviations, but this information is missing) in Tables 1 and 2 are quite large, in many cases on the same order of magnitude as the estimate of the mean AUDC. Taking Table 1 as an example, the top three contenders for Fashion MNIST are MMBS (0.051 ± 0.052), MMBS + SG (0.054 ± 0.049), and GIG (Saliency) + SG (0.078 ± 0.067). Assuming that the stated uncertainties are standard deviations or errors, these results are statistically almost indistinguishable. The same can be said for many of the other methods and datasets."
> > *Reply:* The standard deviations in the AUDC scores do not indicate whether the average AUDC scores of different methods are statistically indistinguishable or not. If you could take infinite samples, you could fully sample the AUDC score distributions, so you would know for certain which method performs better on average, but the standard deviation wouldn’t be zero. The average AUDC score of MMBS was the lowest out of 11 methods, on three different networks, and calculated over 1000 images for each network, which we believe is strong evidence for our claim that MMBS provides state-of-the-art AUDC scores on image classification networks.

---

### Comment · Action_Editor_fj8L · 2026-06-03
**Engage in review**

Dear reviewers,
the authors have now responded to your reviews and it is time to engage with them in any subsequent discussion.

In particularly, I advise you to check not only your part of the review but also the the comments of the other reviewers and the related responses. These may give you a better feeling for the paper and may be particularly useful for reaching your final recommendation.

Thanks
Your AE

---

### Decision · Action_Editor_fj8L · 2026-06-25

**Recommendation:** Accept as is

**Audience:**

Yes

**Audience Explanation:**

The paper proposes a new method for explanations of image classification neural networks. It extends the rather popular attribution method of integrated gradients and shows promising empirical results. All reviewers agree that the method is an interesting contribution to the field of XAI and is likely to trigger interest of researchers in this direction.

**Claims And Evidence:**

Yes

**Claims Explanation:**

While some of the claims in the initial version were not very well supported or were not completely clearly formulated, these points have been raised but the reviewers and addressed by the authors in the updated version of the paper. Reviewers generally appreciated these updates and confirmed the improved quality of the revised version.